



# Dimethyl sulfide chemistry over the industrial era: comparison of key oxidation mechanisms and long-term observations

Ursula A. Jongebloed[1], Jacob I. Chalif[2], Linia Tashmim[3], William C. Porter[3], Kelvin H. Bates[4], Qianjie Chen[5], Erich C. Osterberg[2], Bess G. Koffman[6], Jihong Cole-Dai[7], Dominic A. Winski[8], David G. Ferris[2], Karl J. Kreutz[8], Cameron P. Wake[9], and Becky Alexander[1]

[1]Department of Atmospheric and Climate Science, University of Washington, Seattle, WA, 98195, United States of America
[2]Department of Earth Sciences, Dartmouth College, Hanover, NH, 03755, United States of America
[3]Department of Environmental Sciences, University of California, Riverside, CA, 92521, United States of America
[4]Department of Mechanical Engineering, University of Colorado, Boulder, CO, 80309, United States of America
[5]Department of Civil and Environmental Engineering, The Hong Kong Polytechnic University, Hong Kong SAR 999077, China
[6]Department of Geology, Colby College, Waterville, ME, 04901, United States of America
[7]Department of Chemistry and Biochemistry, South Dakota State University, Brookings, SD, 57006, United States of America
[8]Climate Change Institute and School of Earth and Climate Science, University of Maine, Orono, ME, 04469, United States of America
[9]The Center for North Atlantic Studies, University of New England, Biddeford, ME, 04005, United States of America

**Correspondence:** Ursula A. Jongebloed (ujongebl@uw.edu) and Becky Alexander (beckya@uw.edu)

**Abstract.** Dimethyl sulfide (DMS) is primarily emitted by marine phytoplankton and oxidized in the atmosphere to form methanesulfonic acid (MSA) and sulfate aerosols, which affect climate by scattering incoming solar radiation and influencing cloud properties. Ice cores in regions affected by anthropogenic pollution show an industrial-era decline in MSA, which has previously been interpreted as indicating a decline in phytoplankton abundance. However, a simultaneous increase in DMS-
5 derived sulfate (bioSO$_4$) in a Greenland ice core suggests that pollution-driven oxidant changes caused the decline in MSA by influencing the relative production of MSA versus bioSO$_4$. Here we use GEOS-Chem, a global chemical transport model, over three time periods (preindustrial, peak North Atlantic NO$_x$ pollution, and 21$^{st}$ century) to investigate the chemical drivers of the industrial-era changes in MSA and bioSO$_4$, and examine whether four DMS oxidation mechanisms reproduce trends and seasonality in DMS, MSA, and bioSO$_4$ observations. We find that GEOS-Chem and box model simulations can reproduce
ice core trends in MSA and bioSO$_4$, but model results are sensitive to DMS oxidation mechanism and oxidant concentrations. Our simulations support the hypothesized nitrate-radical driven decline in MSA over the industrial era, but none of the GEOS-Chem simulations can capture the seasonality of in situ DMS observations while simultaneously reproducing trends in ice core MSA and bioSO$_4$. To reduce uncertainty in modeling DMS-derived aerosols, future work should investigate aqueous-phase chemistry, which produces 82–99% of MSA and bioSO$_4$ in our simulations, and constrain atmospheric oxidant concentrations,
including the nitrate radical, hydroxyl radical, and reactive halogens.





## 1 Introduction

Marine phytoplankton are primary producers and an important source of atmospheric sulfur through the emission of dimethyl sulfide (DMS, $CH_3SCH_3$). In the atmosphere, DMS oxidation forms methanesulfonic acid (MSA, $CH_3SO_3H$) and sulfate ($SO_4^{2-}$) aerosols, both of which play an important role in the formation and growth of new particles and cloud condensation nuclei (e.g., Beck et al., 2021; Chen et al., 2015; Weber et al., 1997; Kaufman and Tanré, 1994) and influence aerosol radiative forcing (e.g., Fung et al., 2022; Carslaw et al., 2013; Regayre et al., 2020). Uncertainty in past, present, and future DMS emissions and oxidation chemistry contribute to uncertainty in aerosol radiative forcing estimates (e.g., Carslaw et al., 2013, 2017; Fung et al., 2022; Kaufman and Tanré, 1994). Ice core records of MSA concentrations, traditionally considered a proxy for DMS emissions, have been used to infer phytoplankton abundance (Kurosaki et al., 2022; Osman et al., 2019; Polashenski et al., 2018) and sea ice extent (Abram et al., 2013; Curran et al., 2003; Maselli et al., 2017; Osterberg et al., 2015). Based on industrial-era declines in MSA concentrations across many Greenland ice cores, it was inferred that DMS emissions—and consequently, marine phytoplankton abundance—had declined in the North Atlantic between the preindustrial and early 21st century (Osman et al., 2019). A more recent study found an increase in Greenland MSA from 2002–2014 and attributed the increase to declining sea ice extent (Kurosaki et al., 2022). More recently, sulfur isotopes of sulfate ($\delta^{34}S(SO_4^{2-})$) from a Summit, Greenland ice core showed that DMS-derived sulfate ($bioSO_4$) had increased in the North Atlantic region since the preindustrial (Jongebloed et al., 2023a). The time period of minimum MSA concentrations (1969–1995 CE) aligns with peak anthropogenic $NO_x$ pollution in the regions affecting Greenland, causing Jongebloed et al. (2023a) to hypothesize that the trends in MSA and $bioSO_4$ are driven by changes in DMS oxidation chemistry due to changes in atmospheric oxidant abundances. In support of this hypothesis, a mid-20[th] century through early 21[st]-century decline in MSA concentrations in the Denali, Alaska ice core, which is influenced by DMS emissions from the North Pacific, was found to align with an increase in East Asian oxidant precursor emissions starting in the 1950s (Chalif et al., 2024).

Jongebloed et al. (2023a) and Chalif et al. (2024) hypothesized that increased industrial-era $NO_x$ and VOC emissions drive increases in the nitrate radical ($NO_3$), and that oxidation of DMS by the nitrate radical favors the production of sulfate over the production of MSA. Using a global chemistry-climate model with updated DMS oxidation chemistry, Fung et al. (2022) found a 59% decrease in global MSA burden from the preindustrial to present day in a global climate model with updated DMS oxidation chemistry, supporting the hypothesis of a pollution-driven decline in MSA. Chalif et al. (2024) used a box model with gas-phase DMS oxidation chemistry from recent studies (Fung et al., 2022; Chen et al., 2018; Cala et al., 2023; Novak et al., 2021), and found a $NO_3$-driven decline in modeled MSA concentrations of the same magnitude as the decline in MSA observed in the Summit, Greenland and Denali, Alaska ice cores. However, this box model approach does not include aqueous-phase chemistry, which is likely the dominant MSA formation pathway in the atmosphere (Chen et al., 2018). Additionally, the rapidly evolving representation of DMS oxidation mechanisms in atmospheric chemistry models compels a careful comparison of these various mechanisms.

Many atmospheric models have simple DMS oxidation schemes which include three gas-phase reactions with the hydroxyl radical (OH) and the nitrate radical (Chin et al., 1996). In recent years, these have been updated to include both additional gas-



phase and aqueous-phase reactions involving additional MSA and sulfate precursors. Chen et al. (2018) implemented updates to DMS oxidation chemistry in the global chemical transport model GEOS-Chem, including the reaction with bromine monoxide (BrO) and the chlorine radical (Cl), formation of important intermediates such as dimethyl sulfoxide (DMSO, $CH_3SOCH_3$) and methanesulfinic acid (MSIA, $CH_3SO_2H$), and the aqueous-phase formation of MSA from these intermediates. Tashmim et al. (2024) built on the Chen et al. (2018) mechanism to include gas-phase chemistry producing hydroperoxymethyl thioformate (HPMTF, $HOOCH_2SCHO$), which has been observed in the atmosphere (Novak et al., 2021; Siegel et al., 2023; Veres et al., 2020) and laboratory studies (Goss and Kroll, 2024; Shen et al., 2022; Ye et al., 2022) and forms sulfate in the aqueous phase (Novak et al., 2021). Along with HPMTF, Tashmim et al. (2024) included other gas-phase intermediates such as the methylthiomethylperoxy radical (MSP or MTMP; $CH_3SCH_2OO$) and the $CH_3SO_2$ radical. Chen et al. (2023) implemented a DMS oxidation mechanism in GEOS-Chem that included the temperature-dependent gas-phase production of MSA and sulfate through the $CH_3SO_2$ radical, which has been observed in recent chamber studies (Berndt et al., 2023; Goss and Kroll, 2024; Shen et al., 2022; Ye et al., 2022). Gas-phase production of MSA increases in simulations using the Chen et al. (2023) mechanism relative to the simple three gas-phase reaction mechanism, which could be important for new particle formation.

Similar DMS oxidation mechanisms with a wide range in complexity have been implemented into chemical transport models, chemistry-climate models, and box models. Hoffmann et al. (2021) and Revell et al. (2019) found that different DMS oxidation schemes yield order-of-magnitude differences in $SO_2$, MSA, and sulfate concentrations. Similarly, Fung et al. (2022) found that updating the DMS chemistry mechanism in a chemistry-climate model causes a decreased estimated aerosol radiative forcing, demonstrating the importance of DMS chemistry to climate modeling. Cala et al. (2023) implemented gas-phase DMS oxidation similar to Tashmim et al. (2024) and Fung et al. (2022), but do not include aqueous-phase oxidation of DMS, DMSO, and MSIA or the reaction of DMS with BrO and Cl. Cala et al. (2023) also found significant variation in DMS oxidation products under different oxidation mechanisms and highlight the need to investigate the kinetics of small sulfur radical intermediates ($CH_3S$, $CH_3SO_2$, and $CH_3SO_3$). Finally, Bhatti et al. (2024) used a global chemistry-climate model to implement seven simple DMS oxidation mechanisms from other models, none of which included HPMTF chemistry, and found a range in global aerosol optical depth that is twice as large as the modeled change from preindustrial to present-day aerosol optical depth.

Recent chamber, modeling, and observation studies highlight remaining uncertainties in DMS oxidation, including uncertainty in the MSP isomerization rate to form HPMTF (Jernigan et al., 2022; Wu et al., 2015; Ye et al., 2022); the fate of HPMTF, including gas-phase oxidation to form $SO_2$ (Novak et al., 2021; Jonge et al., 2021; Wu et al., 2015), in-cloud oxidation to form sulfate (Novak et al., 2021; Vermeuel et al., 2020), or photolysis (Khan et al., 2021); the formation and loss of dimethyl sulfone (Scholz et al., 2023; Shen et al., 2022); the kinetics of small sulfur radical intermediates ($CH_3S$, $CH_3SO_2$, and $CH_3SO_3$) to form sulfate and MSA (Berndt et al., 2023; Cala et al., 2023; Chen et al., 2021, 2023; Goss and Kroll, 2024; Ye et al., 2022); the reaction of MSIA in the aqueous phase (Liu et al., 2023); and the reaction of MSA with OH in the aqueous phase to form sulfate (Kwong et al., 2018; Mungall et al., 2018). In addition to remaining uncertainties in DMS oxidation chemistry, uncertainties in modeled oxidant abundances affect the relative abundance of DMS oxidation products, and representation of atmospheric oxidants varies drastically by model (Murray et al., 2021; Young et al., 2013).



Here we implement four DMS oxidation mechanisms from previous studies (Chen et al., 2018; Tashmim et al., 2024; Chen et al., 2023; Cala et al., 2023) into a global atmospheric chemistry model to investigate how modeled abundances of DMS oxidation products over the industrial era compare to long-term in situ observations and to ice cores from Summit, Greenland and Denali, Alaska. We use these four mechanisms to represent a range in the complexity and characteristics of the representation of intermediates. We investigate which oxidants and reactions drive trends in MSA and bioSO$_4$, where knowledge gaps remain in DMS oxidation chemistry, and the potential global implications of these mechanisms for DMS oxidation products.

## 2 Methodology

### 2.1 GEOS-Chem model

To investigate modeled industrial-era trends in MSA and DMS-derived biogenic sulfate, we use GEOS-Chem versions 12.9.3 (abbreviated as GC12; https://zenodo.org/records/3974569) and 13.2.1 (abbreviated as GC13; https://doi.org/10.5281/zenodo.5500717) (Bey et al., 2001). We use two model versions to test the sensitivity of our results to different oxidant concentrations (see Section 3.1). GEOS-Chem is driven by assimilated meteorology from MERRA2 and has detailed HO$_x$-NO$_x$-VOC-O$_3$-halogen chemistry including recently updated halogen and cloud chemistry (Bates and Jacob, 2019; Chen et al., 2017; Holmes et al., 2019; Schmidt et al., 2016; Wang et al., 2019, 2021). We run simulations at $4° \times 5°$ resolution with varying anthropogenic emissions (for years 1750, peak NO$_x$ pollution in 1979, and 2007) from the Community Emissions Data System (CEDS; McDuffie et al., 2020). DMS emissions from the ocean are described in Lana et al. (2011) and are based on a climatology of sea-surface DMS concentrations with flux controlled by sea surface temperature- and wind-dependent gas transfer velocity (Johnson, 2010; Nightingale et al., 2000). In all mechanisms, we add tracers to GEOS-Chem to track DMS-derived SO$_2$ and bioSO$_4$ separately from other sources of SO$_2$ and sulfate while preserving total modeled sulfur. Dry deposition is parameterized as a resistance-in-series model (Wang et al., 1998; Wesely, 1989) and wet deposition includes both scavenging and washout of soluble species (Liu et al., 2001). In version 13.2.1, the wet deposition is updated to include spatially and temporally varying in-cloud condensed water and a higher washout rate for nitric acid (Luo et al., 2019, 2020). To test different time periods, we use the same meteorology and natural emissions from 2007 across all simulations, but prescribe anthropogenic emissions from other years representing each time period, following Zhai et al. (2021) and Jongebloed et al. (2023c).

We implement DMS oxidation chemistry from Chen et al. (2018), Chen et al. (2023), Cala et al. (2023), and Tashmim et al. (2024) to represent a range in DMS oxidation chemistry, such as the inclusion of HPMTF chemistry (Cala, Tashmim, J. Chen), the inclusion of DMS loss to reactive halogens (Q. Chen, Tashmim), and different representations of the short-lived organosulfur intermediates such as CH$_3$SO$_2$ (Cala, Tashmim, and J. Chen). We then quantify global implications of different DMS oxidation mechanisms and oxidant concentrations for DMS oxidation products. DMS oxidation mechanisms are described in Section 2.2.

This study does not consider how changes in meteorology might affect long-term trends in MSA and sulfate. Changes in meteorology are potentially important in the Denali ice core, where the snow accumulation rate has increased by a factor of



1.2–2.3 since the preindustrial (Winski et al., 2017; Chalif et al., 2024). Chalif et al. (2024) showed that these accumulation rate changes alone cannot explain the trend in MSA concentrations, and here we use the same meteorology across all simulations
to investigate how changing atmospheric chemistry influences trends in ice core concentrations of DMS oxidation products.

Importantly, this study also does not consider how potential past and future changes in DMS emissions might affect long-term trends in MSA and sulfate. We use the same DMS emissions from Lana et al. (2011) in every simulation; however, uncertainty in or changes to DMS emissions could affect comparison of model simulations with long-term ice core and in situ observations. DMS emissions inventories in models vary by up to a factor of 2 or more, and may not capture spatiotemporal
variability in DMS emissions (Bhatti et al., 2023; Galí et al., 2018; Hulswar et al., 2022; Lana et al., 2011; Steiner et al., 2012). Furthermore, DMS emissions vary under different temperature, pH, and nutrient availability, and may change under global warming (Hopkins et al., 2020, 2023; Kloster et al., 2007; Saint-Macary et al., 2021; Øyvind Seland et al., 2020; Six et al., 2013; Sunda et al., 2007; Tjiputra et al., 2020; Xu et al., 2022; Zhao et al., 2024; Zindler et al., 2014). Therefore, potential changes in present, past, and future DMS emissions should also be considered when interpreting observed or modeled long-
term trends of MSA and sulfate.

Finally, this study does not include emissions of methanethiol (CH$_3$SH; MeSH), which is emitted at about 3–40% of the rate of DMS emission (Gros et al., 2023; Lawson et al., 2020; Novak et al., 2022), and may favor SO$_2$ and sulfate production over MSA (Novak et al., 2022), potentially affecting our model-observation comparison of MSA, bioSO$_4$, and MSA/bioSO$_4$.

### 2.2 DMS oxidation mechanisms

We perform simulations using four DMS oxidation mechanisms, which are summarized in Table 1 and Figure 1, and discussed in detail below. Detailed schematics for the mechanisms can be found in Figures S1-S4. Table S1 shows Henry's law constants of aqueous-phase intermediates for the four mechanisms and Table 2 shows the time periods simulated.

In the Q. Chen mechanism, we use the DMS oxidation scheme from Chen et al. (2018). This mechanism includes DMS oxidation by key oxidants, including OH via addition and abstraction, the nitrate radical, bromine monoxide (BrO), ozone (O$_3$)
via gas- and aqueous-phase chemistry, and the chlorine radical (Cl). Chen et al. (2018) include aqueous-phase production of MSA via DMSO reacting with OH to produce MSIA, followed by MSIA reacting with OH and ozone to produce MSA. Chen et al. (2018) also include the aqueous-phase destruction of MSA by OH to produce sulfate. However, this reaction appears to be overly efficient when implemented in the Tashmim mechanism (Fig. S5) and is therefore omitted from all mechanisms in this study.

The Tashmim mechanism includes all the reactions in the aqueous-phase and addition pathways from Chen et al. (2018) (Tashmim et al., 2024). This mechanism updates the abstraction pathway to include the isomerization component, including intermediates such as MSP, CH$_3$SO$_2$, HPMTF, and the aqueous-phase formation of sulfate from HPMTF.

The J. Chen mechanism includes the gas-phase chemistry from Chen et al. (2023) and aqueous-phase chemistry from Chen et al. (2018) and Tashmim et al. (2024). Henry's law constants for all aqueous-phase species are from Chen et al. (2023) (see
Table S1). The main difference between the J. Chen mechanism and the Tashmim mechanism is that the J. Chen mechanism adds gas-phase MSA and sulfate production through the CH$_3$SO$_2$ radical intermediate. Other important differences include





**Table 1.** Description of mechanisms.

| Mechanism Name | GEOS-Chem Version | Description [a,b] | Citation(s) |
|---|---|---|---|
| Q. Chen | GC12 | DMS oxidation mechanism from Chen et al. (2018) with MSA + OH (aq) $\rightarrow SO_4^{2-}$ turned off[c]. | Chen et al. (2018) |
| Tashmim | GC12, GC13 | DMS oxidation mechanism from Tashmim et al. (2024) (which has the same aqueous-phase chemistry as the Q. Chen mechanism), and adds gas-phase chemistry in the abstraction pathway including MSP and HPMTF and the aqueous-phase formation of sulfate from HPMTF in cloud and aerosol. | Tashmim et al. (2024), Chen et al. (2018) |
| J. Chen | GC12 | Gas-phase DMS oxidation mechanism from Chen et al. (2023), which adds gas-phase production of MSA and sulfate through the $CH_3SO_2$ radical. Aqueous-phase chemistry from the Q. Chen and Tashmim mechanisms is implemented in this mechanism. This mechanism does not include DMS + $O_3$, DMS + BrO, or DMS + Cl. | Chen et al. (2023) |
| Cala | GC13 | Gas-phase DMS oxidation mechanism from Cala et al. (2023). Aqueous-phase chemistry from the Q. Chen and Tashmim mechanisms is implemented in this mechanism. This mechanism does not include DMS + $O_3$, DMS + BrO, or DMS + Cl. | Cala et al. (2023) |

[a] Detailed schematics of each mechanism are shown in Fig. S1-S4. Reaction rates, and full descriptions of the mechanisms can be found in the citations associated with each mechanism.

[b] In all mechanisms, $SO_2$ is oxidized to sulfate in the gas phase through reaction with OH and in the aqueous phase through reaction of S(IV) with $H_2O_2$, $O_3$, HOBr, HOCl, and $O_2$ catalyzed by transition metals iron and manganese (Alexander et al., 2009, 2012; Chen et al., 2017).

[c] Other sensitivity tests performed include the Tashmim mechanism with MSA + OH (aq) $\rightarrow SO_4^{2-}$ included, shown in Figure S5.

**Table 2.** Time periods simulated in the GEOS-Chem model.

| Time Period | Description |
|---|---|
| 1750 | 2007 meteorology, 2007 natural emissions, and 1750 anthropogenic emissions. |
| 1979 | 2007 meteorology, 2007 natural emissions, and 1979 anthropogenic emissions. |
| 2007 | 2007 meteorology, 2007 natural emissions, and 2007 anthropogenic emissions. |

the omission of the DMS + BrO, DMS + $O_3$, and DMS + Cl reactions. Although Fung et al. (2022), Chen et al. (2018), and Khan et al. (2016) show that DMS + BrO may be a significant sink for DMS (8–29% globally), we omit these reactions from our J. Chen mechanism for consistency with Chen et al. (2023). Other notable differences between the J. Chen and

Tashmim mechanisms include intermediates such as methanesulfenic acid (MSEA), and the reactions connecting the addition



and abstraction branches through the oxidation of MSEA and MSIA to form $CH_3SO_2$, which can then form both MSA and sulfate. Finally, we include the aqueous-phase chemistry from Chen et al. (2018) and Tashmim et al. (2024) in the J. Chen mechanism to include the aqueous-phase formation of MSA and sulfate from DMSO and HPMTF, respectively.

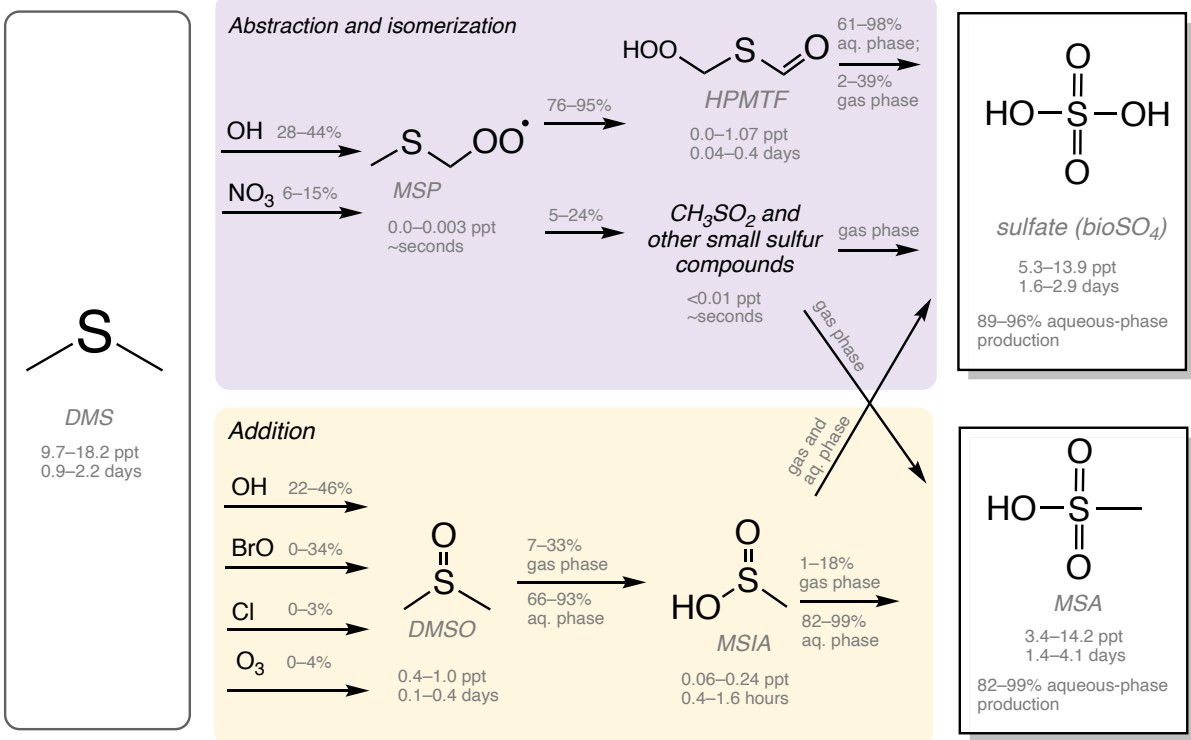

**Figure 1.** Simplified schematic of DMS oxidation chemistry in Table 1, which includes the DMS oxidation mechanisms in Chen et al. (2018), Tashmim et al. (2024), Chen et al. (2023), and Cala et al. (2023). The purple box shows the abstraction and isomerization branch, which forms MSP, HPMTF, $CH_3SO_2$, other short-lived organosulfur compounds, $bioSO_4$, MSA, and carbonyl sulfide (OCS) (Chen et al., 2018; Tashmim et al., 2024; Chen et al., 2023; Cala et al., 2023). In the yellow box, the addition branch includes DMSO, MSIA, $SO_2$, $bioSO_4$, and MSA (Chen et al., 2018; Tashmim et al., 2024). DMSO, MSIA, and HPMTF intermediates partition into the aqueous phase. Percentages above reaction arrows show the percent of the precursor oxidized through each pathway, where the range is based on range in the global mean in simulations with various oxidation mechanisms in Table 1. Below each compound, a range in mass-weighted mean tropospheric concentrations and tropospheric lifetimes is shown as a range across all mechanisms and model versions in Table 1. All concentration, lifetime, and percent oxidation numbers are from the 2007 (present day) simulations. $SO_2$ produced from DMS oxidation can be dry deposited or oxidized in the gas and aqueous phases to form sulfate (Alexander et al., 2009, 2012; Chen et al., 2017). Reaction rates and detailed descriptions of the reactions can be found in Chen et al. (2018), Tashmim et al. (2024), Chen et al. (2023), and Cala et al. (2023).

For the Cala mechanism, we implement gas-phase DMS oxidation chemistry from Cala et al. (2023) with aqueous-phase chemistry from Chen et al. (2018) and Tashmim et al. (2024). We note that oxidation of HPMTF in aerosol to form sulfate is similar in Cala et al. (2023) and Tashmim et al. (2024), but Cala et al. (2023) do not include aqueous-phase oxidation of





DMSO and MSIA or cloud loss of HPMTF. Similar to the J. Chen mechanism, the Cala mechanism does not include DMS + BrO, DMS + $O_3$, or DMS + Cl.



**Figure 2.** Ice core observations from Denali, Alaska (left; Chalif et al., 2024) and Summit, Greenland (right; Jongebloed et al., 2023a). a) Denali MSA concentrations, where the thin line is annual concentrations and the thick line is annual concentrations smoothed with a Hann window function. b) Summit MSA concentrations, which are sub-decadal from 1700 to 1980 and annual from 1980 to 2007. c) a map showing the locations of Denali (63°N, 151°W) and Summit (73°N, 39°W) and the source regions shown as dashed lines for Denali (Polashenski et al., 2018) and Summit (Osman et al., 2019). d) Summit bioSO$_4$ concentrations. e) Summit MSA/bioSO$_4$ molar ratios. Dotted lines show Bayesian changepoint analysis from Jongebloed et al. (2023a) and Chalif et al. (2024). Error bars from Summit core (b, d, and e) show 1-sigma propagation of uncertainty in isotope measurements and calculations.



## 2.3 Box modeling of DMS oxidation chemistry

We use the Framework for 0-Dimensional Atmospheric Modeling (F0AM, Wolfe et al., 2016) to isolate the impacts of changing oxidant concentrations on trends in MSA and bioSO$_4$. We follow Chalif et al. (2024) by using March–October mass-weighted oxidant concentrations in the marine boundary layer (<2 km) from the 1750, 1979, and 2007 simulations (Table 1) as inputs for the box model. We model the four oxidation mechanisms described in Table 1 and oxidant concentrations from both GC12 and GC13. Box model simulations, including those in Chalif et al. (2024), only model gas-phase chemistry. We approximate the aqueous-phase pathways by allowing MSIA oxidation to form only MSA, which is informed by GEOS-Chem simulations where 90% of the MSIA forms MSA (Fig. 1). This approximation of aqueous-phase chemistry increases the absolute ratio of MSA/bioSO$_4$ by about a factor of two, but does not affect modeled trends in MSA, bioSO$_4$, or MSA/bioSO$_4$ in any mechanism.

## 2.4 Long-term observations: ice core and in situ measurements

We use ice core measurements of MSA from the Denali, Alaska ice core (Chalif et al., 2024) and MSA and bioSO$_4$ from a Summit, Greenland ice core (Jongebloed et al., 2023a) to investigate observed trends in DMS oxidation products. Figure 2 shows ice core observations and the locations of Denali and Summit. The Denali ice core is located in the sub-Arctic North Pacific region, which is influenced by East Asian emissions, and includes annually-resolved MSA concentrations from 1700 to 2013 CE (Fig. 2a). The Summit ice core is from the sub-Arctic North Atlantic region, which is influenced by anthropogenic emissions from eastern North America and western Europe, includes MSA concentrations (Fig. 2b) that are consistent with a composite MSA record from ice cores across Greenland (Fig. S6; Osman et al., 2019). The Summit ice core observations also include DMS-derived biogenic sulfate (bioSO$_4$) concentrations (Fig. 2d) and the ratio of MSA/bioSO$_4$ from 1700 to 2007 (Fig. 2e) determined via isotope apportionment of sulfate sources (Jongebloed et al., 2023a, b, c). We cannot estimate bioSO$_4$ and MSA/bioSO$_4$ for the Denali ice core because the sulfur isotopic composition of sulfate was not measured.

To compare the model results to the ice core observations, we have considered several methods based on previous work. Zhai et al. (2021) and Jongebloed et al. (2023c) compared Greenland ice cores to the tropospheric burden of relevant species in a 5-day HYSPLIT back-trajectory region of Greenland. Osman et al. (2019) and Chalif et al. (2024) investigated Greenland ice cores using a smaller HYSPLIT back-trajectory region. Moseid et al. (2022) and Zhang et al. (2024) compared sulfate and black carbon in ice cores to several models by examining the modeled deposition of each species in the grid cell containing the ice core. For this study, we follow Moseid et al. (2022) and Zhang et al. (2024) and compare the trends in grid cell deposition to trends in ice core concentration, but results are qualitatively similar using other methods.

In addition to ice core observations, we compare model results to long-term in situ observations of MSA, DMS, and MSA/nssSO$_4^{2-}$ from Ayers et al. (1995), Becagli et al. (2019), Gondwe et al. (2004), Kouvarakis and Mihalopoulos (2002), Quinn et al. (2009), Schmale et al. (2022), and Sharma et al. (2019). We include details on these in situ measurements and note the limitations of these comparisons in Section 3.4.





## 3    Results and discussion

### 3.1    Changes in oxidant concentrations across model versions and simulation years

Figure 3 shows that concentrations of DMS oxidants are substantially different across two different GEOS-Chem model versions (GC12 and GC13) and simulation years (1750, 1979, and 2007) in the Summit source region (Fig. 3a-e), Denali source region (Fig. 3f-j), and global mean (Fig. 3k-o). We analyzed the Summit and Denali source regions (Fig. 3a and 3f) to examine how oxidants have changed in the upwind regions influencing each ice core site (Osman et al., 2019; Polashenski et al., 2018; Chalif et al., 2024). Oxidants are influenced by trends in anthropogenic pollution, which differ across each of these regions over the industrial era. We show $NO_x$ emissions from the Community Emissions Data System (CEDS; McDuffie et al., 2020) in Figure 3, which increase from the late 19[th] through late 20[th] century and decrease from the late 20[th] through early 21[st] century in North America and the European Union (i.e., upwind of Summit; Fig. 3b-e). In East Asia, (i.e., upwind of Denali; Fig. 3g-j), $NO_x$ emissions increase rapidly in the late 20[th] through early 21[st] century. We show concentrations of OH, $NO_3$, BrO, and $O_3$, but not the chlorine radical (Cl), because modeled trends in Cl are similar to results shown in Zhai et al. (2021) and Cl is expected to be a minor oxidant of DMS; however, we note that anthropogenic emissions of chlorine are not included in either model version and current reactive chlorine chemistry mechanisms underestimate observed reactive chlorine (Chen et al., 2022, 2024).

Tropospheric nitrate radical concentrations increased between 1750 and 1979 and plateaued between 1979 and 2007 in the Summit source region (Fig. 3b) and Denali source region (Fig. 3g) and increased further in the global mean (Fig. 3l). The tropospheric nitrate radical concentrations increased by similar factors in both GC12 and GC13 in the Summit source region (factor of 2.3 and 2.8), Denali source region (1.2 and 1.6), and global mean (0.7 and 1.0), consistent with changes simulated by another global model (Khan et al., 2015). Notably, the absolute concentrations are a factor of 2.2–4.5 higher in GC13 than GC12. The higher nitrate radical concentrations may be driven in part by differences in ozone concentrations in model versions (Fig. 3d, 3i, and 3n), which are consistently 6–11 ppb higher in GC13 compared to version 12.9.3 across all simulation years. Both versions are within the range of ozone concentrations over the industrial era in Atmospheric Chemistry and Climate Model Intercomparison Project (ACCMIP) simulations (Young et al., 2013). In contrast to ozone and the nitrate radical, BrO concentrations are 0.04–0.19 ppt lower in GC13 compared to GC12. Modeled concentrations of BrO in the Summit source region increased by 12–42% from 1750 to 2007 (Fig. 3b), similar to a 16% increase in Russian Arctic BrO concentrations modeled by Zhai et al. (2024). Tropospheric mean OH is up to $1.7 \times 10^5$ molec cm$^{-3}$ higher in 13.2.1 compared to 12.9.3, but changes between 1750 and 1979 mean OH vary from $-19\%$ to $+13\%$ (Fig. 3e, 3j, and 3o). The 2–8% increase in OH between 1979 and 2007 is consistent with modeled northern hemisphere OH anomalies based on methane $^{13}C$ (Turner et al., 2017) and trends in global mean tropospheric OH from ACCMIP simulations (Murray et al., 2021). Investigating the reasons for the differences in oxidant concentrations between model versions is beyond the scope of this study, but the substantial differences in modeled oxidant concentrations between GC12 and GC13 are useful for examining the sensitivity of DMS oxidation to oxidant concentrations.





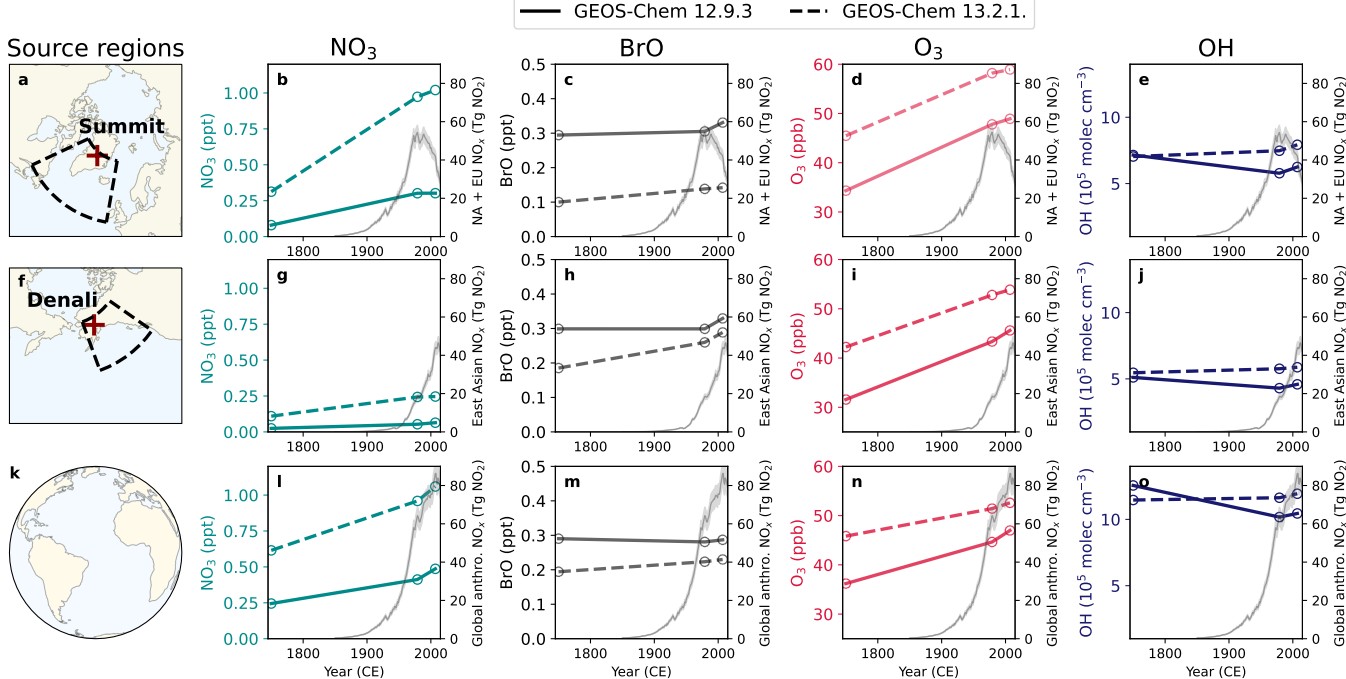

**Figure 3.** Annual air-mass weighted tropospheric mean oxidant concentrations in the Summit source region (top), Denali source region (middle), and global mean from the preindustrial to the present day. In all figures showing oxidants, the dashed lines represent oxidant concentrations from version GEOS-Chem 13.2.1 (GC13) and the solid lines represent GEOS-Chem 12.9.3 (GC12). CEDS anthropogenic $NO_x$ emissions from North America and the European Union (b-e), East Asia (g-j) and global (l-o) are shown with a gray line with shading showing one standard deviation (McDuffie et al., 2020). Meteorology and natural emissions are the same in all simulations. a) The Summit ice core site and source region. b-e) Changes in tropospheric air-mass weighted $NO_3$ (turquoise), BrO (gray), $O_3$ (red), and OH (dark blue) in the 1750, 1979, and 2007 simulations in the Summit ice core source region. f) The Denali ice core site and source region. g-j) Changes in tropospheric air-mass weighted $NO_3$ (turquoise), BrO (gray), $O_3$ (red), and OH (dark blue) in the 1750, 1979, and 2007 simulations in the Denali ice core source region. k) An icon representing the global calculations in l-o. l-o) Global tropospheric air-mass weighted $NO_3$ (turquoise), BrO (gray), $O_3$ (red), and OH (dark blue) in the 1750, 1979, and 2007 simulations.

## 3.2 Comparison between ice core and modeled changes in MSA, bioSO$_4$, and MSA/bioSO$_4$

Figure 4 shows that some GEOS-Chem and box model simulations partially capture Summit trends in MSA, bioSO$_4$, and
MSA/bioSO$_4$, and box model simulations partially capture Denali MSA trends, but all GEOS-Chem simulations model MSA trends that are opposite to the observed trends at Denali. At Denali, ice core MSA concentrations decline by $32 \pm 13\%$ between the preindustrial (1700–1962; Chalif et al., 2024) and late 20$^{th}$ century (1962-1995), and by $49 \pm 13\%$ between the preindustrial and the turn of the century (1996 to 2013). In contrast, the modeled MSA is 7–31% higher in the Denali grid cell in 1979 and 2007 compared to 1750 in all mechanisms and model versions in GEOS-Chem (Fig. 4a). All box model simulations produce



a decline in MSA from 1750 to 1979 and a small increase from 1979 to 2007. The reasons for the discrepancy between GEOS-Chem simulations and observations from Denali are explored in Section 3.3.

Some model simulations partially capture observed trends in Summit MSA (Fig. 4b). Summit ice core MSA concentrations decrease by $57 \pm 19\%$ between the preindustrial (1200 to 1865; Jongebloed et al., 2023a) and Greenland minimum MSA concentrations (1969 to 1995). At the top of the ice core (1996–2007), MSA increases to be only $31 \pm 17\%$ lower relative to the preindustrial. The Cala and Tashmim mechanisms using GC13 oxidants in the box model reproduce the direction of these trends, but are too small in magnitude. In GEOS-Chem, the Cala and Tashmim mechanisms in GC13 and the Chen mechanism in GC12 simulate a decrease in MSA across these time periods of 16–36%, which is also qualitatively similar to the ice core trend but smaller in magnitude (Fig. 4b). None of the GEOS-Chem simulations show an increase in MSA between 1979 and 2007, in contrast to the observed $59 \pm 29\%$ increase in the Summit ice core. In addition, the Tashmim mechanism in GC12 produces trends in MSA from 1750 to 1979 and 2007 (12–14% increase) that are opposite in sign to results produced by the Tashmim mechanism in GC13 (35–36% decrease), indicating sensitivity of these results to oxidants in different model versions.

Summit ice core $bioSO_4$ increases from the preindustrial to Greenland MSA minimum (1969 to 1995) by $20 \pm 11\%$ and decreases to the preindustrial mean at the turn of the century (1996 to 2007; Fig. 4d). However, no GEOS-Chem simulations show a significant increase in $bioSO_4$ in the Summit grid cell. Across all GEOS-Chem simulations, the change in $bioSO_4$ from 1750 to 1979 and from 1750 to 2007 ranges from $-33$ to $+1\%$. In contrast, the box model simulations with the Cala, Tashmim, and J. Chen mechanisms show an increase in $bioSO_4$, qualitatively aligning with the ice core trends. Sulfur isotopes of sulfate have not been measured in the Denali ice core, so we cannot estimate $bioSO_4$ concentrations, but all modeled mechanisms and model versions simulate an increase in $bioSO_4$ of 6–87% between 1750 and 1979 to 2007 (Fig. 4c).

Figure 4f shows that modeled $MSA/bioSO_4$ at Summit varies from 0.2 to 2.0 across model versions and DMS oxidation mechanisms. In comparison, ice core $MSA/bioSO_4$ ranges from 0.06 to 0.56 (Fig. 1e; Jongebloed et al., 2023a). Summit ice core $MSA/bioSO_4$ decreases from $0.25 \pm 0.09$ in the preindustrial to $0.09 \pm 0.04$ during the MSA minimum and increases to $0.15 \pm 0.07$ at the top of the ice core. The box model using the Cala and Tashmim mechanisms and GC13 oxidants qualitatively align with ice core trends in $MSA/bioSO_4$. In GEOS-Chem, the Cala mechanism in GC13 simulates a decrease in $MSA/bioSO_4$ similar to the observed ice core decrease, but $MSA/bioSO_4$ is a factor of 2.4–5.3 higher than ice core $MSA/bioSO_4$ over these time periods (Fig. 4f). The J. Chen mechanism in GC12 also simulates a decrease in $MSA/bioSO_4$ from 1750 to 1979 of $29 \pm 11\%$, which is the largest of all simulations, but still smaller than the observed decrease in $MSA/bioSO_4$ of $60 \pm 36\%$ in the Summit ice core. The J. Chen mechanism also shows the highest $MSA/bioSO_4$ ratios in both GEOS-Chem and box model simulations of up to an order of magnitude higher than the ice core $MSA/bioSO_4$. The Tashmim mechanism in GC13 simulates $MSA/bioSO_4$ of 0.20–0.22 in the Summit grid cell, which is within the range of observed $MSA/bioSO_4$; however, unlike the Summit ice core, modeled $MSA/bioSO_4$ show negligible changes between 1750, 1979, and 2007 in both GEOS-Chem and box model simulations. The Tashmim mechanism in GC12 simulates the opposite trends observed in the Summit ice core and substantially different from results using the Tashmim mechanism in GC13 (Fig. 4f). $MSA/bioSO_4$ in the Denali ice core is not available, but the range in $MSA/bioSO_4$ in the Denali grid cell is similar to the range at Summit.



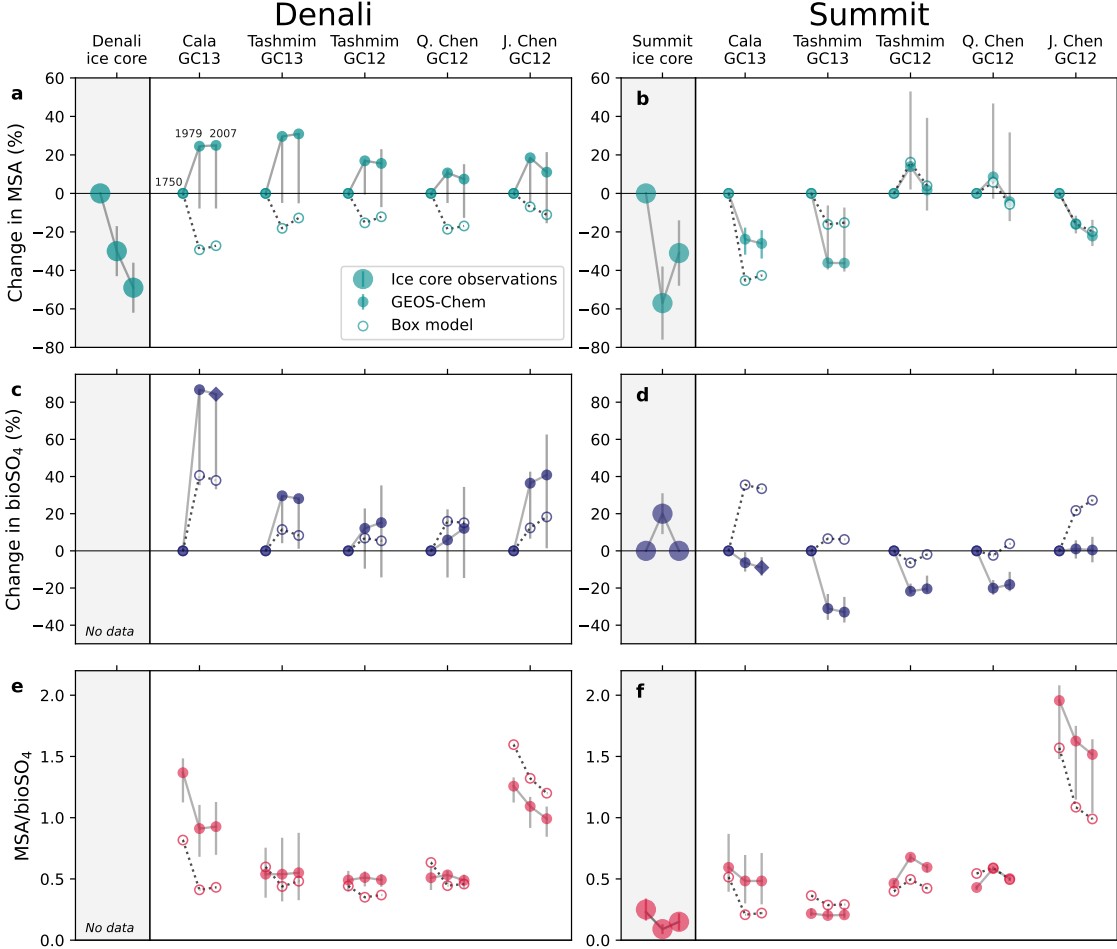

**Figure 4.** Ice core MSA (top), bioSO$_4$ (middle), and MSA/bioSO$_4$ (bottom) in the Denali (left) and Summit (right) ice cores compared to GEOS-Chem and box model results. Large markers are ice core observations, small solid markers are GEOS-Chem model results, and small outlined markers are box model results. MSA and bioSO$_4$ changes are shown as a percent change from the preindustrial for both ice core observations and model results. Markers show the percent change relative to the 1750 preindustrial baseline (left marker, always zero) in 1979 (middle marker) and 2007 (right marker) for MSA (a-b) and bioSO$_4$ (c-d). In e and f, MSA/bioSO$_4$ is shown for 1750 (left markers), 1979 (middle markers) and 2007 (right markers). Denali ice core observations in a and c are shown as percent changes between the preindustrial (1700 to 1962), the late-20[th] century (1962 to 1995), and top of the ice core (1996 to 2013). Summit ice core observations in b and d are shown as percent changes between the preindustrial (1200 to 1865), Greenland minimum MSA (1969 to 1995), and top of ice core (1996 to top of ice core). Ice core error bars show the uncertainty propagated from measurement error and uncertainty in sulfur isotopic source signatures (Jongebloed et al., 2023a; Chalif et al., 2024). GEOS-Chem model error bars show the range in MSA, bioSO$_4$, and MSA/bioSO$_4$ in the ice core grid cell and the surrounding eight grid cells. DMS emissions are the same in all GEOS-Chem simulations and DMS concentrations are the same in all box model simulations.





In summary, some of the GEOS-Chem and box model simulations reproduce observed trends in the Summit, Greenland ice
core MSA, bioSO$_4$, and MSA/bioSO$_4$, but there are substantial differences across mechanisms and model versions. Box model
simulations that consider only changes in oxidants and DMS chemistry mechanisms are able to at least qualitative reproduce
the observations in some simulations. In GEOS-Chem, only mechanisms without DMS + BrO (Cala and J. Chen) simulate both
a decrease in MSA and a decrease in MSA/bioSO$_4$ in qualitative alignment with the Summit ice core. Mechanisms with DMS
+ BrO (Tashmim and Q. Chen) simulate negligible change or the opposite change as the trends observed in Summit MSA,
bioSO$_4$, and MSA/bioSO$_4$. No GEOS-Chem simulation reproduces the observed increase in Summit bioSO$_4$, but box model
simulations produce more positive bioSO$_4$ trends compared to GEOS-Chem, in better alignment with the Summit ice core. All
but one GEOS-Chem simulations overestimates the MSA/bioSO$_4$ ratio. The Tashmim mechanism simulates different trends in
GC12 compared to GC13 in both GEOS-Chem and box model simulations, indicating the sensitivity of these results to oxidant
concentrations. The differences in results across model versions and oxidation mechanisms are investigated in Section 3.3.

**3.3   Explaining modeled changes in MSA, bioSO$_4$, and MSA/bioSO$_4$**

Figure 5 shows that the DMS oxidation rate by each oxidant changed between the preindustrial and polluted time periods
in every mechanism and model version in the GEOS-Chem simulations. In the Summit source region across all simulations,
oxidation of DMS via OH (addition) and OH (abstraction) decreases from 1750 to 1979 and 2007 by 7–30% (Fig. 5a) due
to the combination of a decrease in OH concentration over these time periods in GC12 (Fig. 3e) and competition from other
oxidants in both GC12 and GC13. Oxidation of DMS by the nitrate radical increases by a factor of 2.8–11 in all simulations
(Fig. 5b) due to an increase in nitrate radical concentrations in all simulations (Fig. 3b). This increase in DMS + NO$_3$ favors
bioSO$_4$ production, leading to a change in MSA/bioSO$_4$ partitioning that drives a decrease in MSA. In the Tashmim (GC13),
Tashmim (GC12) and Q. Chen (GC12) simulations, which are the only simulations that include DMS + BrO and DMS + O$_3$,
an increase in BrO and O$_3$ concentrations (Fig. 3c and 3d) drive a 35–110% increase in DMS + BrO and a 3–18% increase
in DMS + O$_3$. These changes cause an increase in DMS oxidation via the addition pathway, which favors MSA production
and drives an increase in MSA, a decrease in bioSO$_4$, and subsequently an increase in MSA/bioSO$_4$ in Tashmim (GC12) and
J. Chen (GC12) simulations (Fig. 4). Interestingly, the lower BrO concentrations and higher nitrate radical concentrations in
GC13 compared to GC12 cause different results in simulations using the same mechanism. The Tashmim mechanism in GC13
simulates a 5–10% decrease in MSA/bioSO$_4$ from 1750 to 1979 despite the increase in the addition pathway driven by DMS
+ BrO. In contrast, the increase in DMS + BrO dominates in the Tashmim mechanism in GC12, leading to a 48% increase in
MSA/bioSO$_4$ from 1750 to 1979 (Fig. 4f).

DMS oxidation by each pathway also changes over the industrial era in the Denali source region (Fig. 5b). The change in
contributions of the DMS + OH (addition) plus DMS + OH (abstraction) pathways ranges from $-12\%$ to $+14\%$ from 1750 to
1979 and 2007. In the Denali source region, DMS + NO$_3$ increases by a factor of 3.5–21, driving an increase in the abstraction
and isomerization pathway that drives a decrease in MSA/bioSO$_4$ in the Cala (GC13) and J. Chen (GC12) simulations (Fig.
4e). However, in the Tashmim (GC13), Tashmim (GC12) and Q. Chen (GC12) simulations, the up to 120% increase in DMS
+ BrO and up to 30% increase in DMS + O$_3$ drives an increase in DMS oxidation via the addition pathway, favoring MSA





**Figure 5.** Annual tropospheric mean reaction rate ($10^3$ molec cm$^{-3}$ s$^{-1}$) of DMS via OH-addition (blue with hatching), OH-abstraction (blue), nitrate radical (green), BrO (gray), chlorine radical (yellow), and ozone (red) in the a) Summit source region, b) Denali source region, c) and global mean. The model mechanism from Table 1 and GEOS-Chem version are shown on the x-axis. For each mechanism and model version, the left column shows 1750, the middle shows 1979, and the right column shows 2007 (Table 1).





production and offsetting the increase in the abstraction and isomerization pathway, leading to a small increase of 0–5% in MSA/bioSO$_4$ (Fig. 5e).

The changes in DMS oxidation between 1750, 1979, and 2007 are relatively small on the global scale (Fig. 5c) due to a relatively smaller change in global mean oxidant concentrations (Fig. 3l-o). Across all simulations, DMS + OH (addition) is 22–49%, DMS + OH (abstraction) is 36–48%, DMS + BrO is 0–34%, DMS + NO$_3$ is 1–15%, DMS + O$_3$ is 0–4%, and DMS + Cl is 0–3% (Figure 5c). The relative contribution from the nitrate radical increased by 1–15% globally from 1750 to 1979 and 2007, while the reactions between DMS and other oxidants changed by ±5% (Fig. 5c).

Figure 6 shows that a decrease in the atmospheric lifetime of DMS and oxidation intermediates (DMSO, MSIA, SO$_2$, etc.) due to increasing oxidant concentrations can cause a local trend in DMS oxidation products (MSA + bioSO$_4$) in regions affected by anthropogenic pollution. While DMS emissions were the same in the 1750 and 2007 simulations, regional changes in MSA + bioSO$_4$ deposition of up to ±50% occur in regions where pollution affects oxidant concentrations. As anthropogenic pollution causes modeled oxidant concentrations to increase (Fig. 3), the global mean DMS lifetime decreases in all simulations

by 10–19% (Fig. S7). As a result, the deposition of DMS oxidation products (MSA + bioSO$_4$) increases in regions within or near both DMS emissions and oxidant changes, such as the North Atlantic, North Pacific, and the Southern Ocean near South America and Australia, where DMS is oxidized more quickly relative to the preindustrial (Figure 6b). Simultaneously, Figure 6b shows that MSA + bioSO$_4$ deposition decreases in regions that are distant from DMS emissions and influenced by pollution (i.e., over continents such as North America, Eurasia, and North Africa). While Figure 6 shows MSA + bioSO$_4$ for the Cala

simulations, results for other simulations are similar.

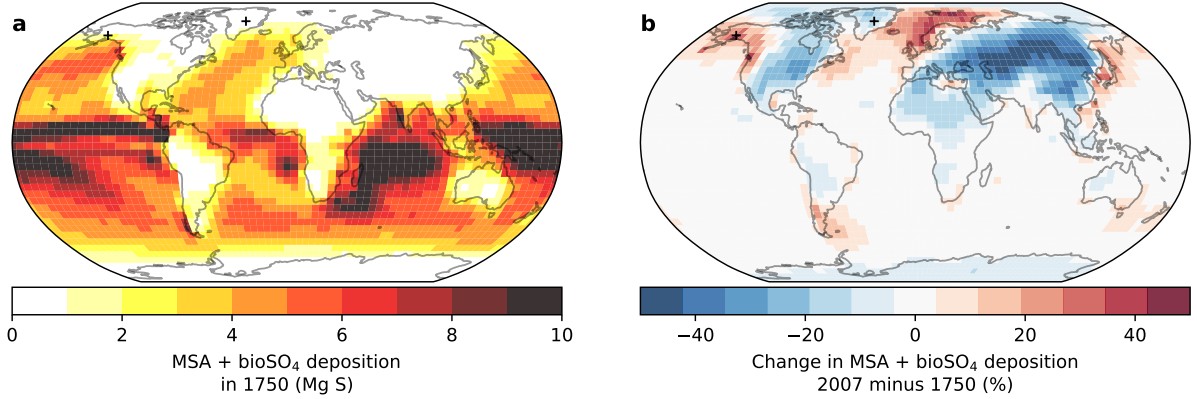

**Figure 6.** Modeled deposition of total biogenic sulfur (MSA + bioSO$_4$) in 1750 (a) and the percent change from 1750 to 2007 (b) in the Cala (GC13) simulations. Markers show the locations of the Denali and Summit ice cores. DMS emissions are the same in both simulations.

The modeled changes in MSA + bioSO$_4$ deposition is smaller in Antarctica compared to Denali and Summit (Fig. 6b) due to the relatively small influence of anthropogenic pollution in this region. We examine modeled trends in MSA, bioSO$_4$, and MSA/bioSO$_4$ in five grid cells containing Antarctic ice core records of MSA concentrations (Abram et al., 2010; Becagli et al., 2009; Curran et al., 2003; Nilsson et al., 2024; Osman et al., 2017; Rahaman et al., 2016; Vega et al., 2016), and find



that the model simulates a change in MSA, bioSO$_4$, and MSA/bioSO$_4$ of $< \pm 10\%$ at any ice core location (Fig. S8). The

relatively low influence of pollution on Southern Ocean region oxidants, and consequently MSA, bioSO$_4$, and MSA/bioSO$_4$,

indicates that trends in MSA may be driven by other factors such as sea ice concentration, primary production, meteorology,

and other variables. However, MSA can undergo post-depositional loss in low-accumulation regions such as East Antarctica

via oxidation to form sulfate, which can affect long-term trends in MSA (Hattori et al., 2024). Additionally, vertical migration

of the methanesulfonate ion can smooth annual and sub-decadal signals, especially in low-accumulation regions (Osman et al.,

2017).

A trend in MSA or bioSO$_4$ due to a decrease in DMS lifetime can offset or amplify a trend in that occurs due to a change

in MSA/bioSO$_4$ partitioning, which is demonstrated in Figure 7. To estimate the effect of changing DMS lifetime on local

trends in MSA deposition in Figure 7, we multiply the fractional change in modeled MSA + bioSO4 deposition (Fig. 6b) by

the preindustrial MSA deposition flux in each grid cell:

$$\Delta\text{MSA}_{\text{DMS lifetime}} = \frac{(\text{MSA} + \text{bioSO}_4)_{1979} - (\text{MSA} + \text{bioSO}_4)_{1750}}{(\text{MSA} + \text{bioSO}_4)_{1750}} \times \text{MSA}_{1750} \tag{1}$$

Where $\Delta\text{MSA}_{\text{DMS lifetime}}$ is the change in MSA deposition due to a change in DMS lifetime between 1750 and 1979, (MSA

+ bioSO$_4$)$_{1979}$ is the MSA + bioSO$_4$ deposition in 1979, (MSA + bioSO$_4$)$_{1750}$ is the MSA + bioSO$_4$ deposition in 1750, and

MSA$_{1750}$ is the MSA deposition in 1750. The change in DMS lifetime also incorporates the change in lifetime of other MSA

and bioSO$_4$ precursors, e.g. DMSO, MSIA, and SO$_2$.

To estimate the effect of changing MSA/bioSO$_4$ partitioning on MSA trends, we subtract $\Delta\text{MSA}_{\text{DMS lifetime}}$ from the change

in MSA:

$$\Delta\text{MSA}_{\text{partitioning}} = \Delta\text{MSA} - \Delta\text{MSA}_{\text{DMS lifetime}} \tag{2}$$

Where $\Delta\text{MSA}_{\text{partitioning}}$ is the change in MSA between 1750 and 2007 due to a change in MSA/bioSO$_4$ partitioning, which is

caused by oxidants favoring different DMS pathways (Fig. 1), and $\Delta\text{MSA}$ is the total change in MSA deposition between 1750

and 1979. Equations 1 and 2 are also applied to bioSO$_4$ to estimate the change in bioSO$_4$ due to a change in DMS lifetime

($\Delta\text{bioSO}_{4\text{DMS lifetime}}$) and change in bioSO$_4$ due to a change in MSA/bioSO$_4$ partitioning ($\Delta\text{bioSO}_{4\text{partitioning}}$).

Figure 7 shows that modeled $\Delta\text{MSA}_{\text{partitioning}}$ and $\Delta\text{bioSO}_{4\text{partitioning}}$ can be offset or amplified by $\Delta\text{MSA}_{\text{DMS lifetime}}$ and

$\Delta\text{bioSO}_{4\text{DMS lifetime}}$. While the Denali ice core shows a change in MSA concentration of $-32 \pm 13\%$ between the preindustrial

and 1962–1995, the model simulates 7–51% increase in MSA due to $\Delta\text{MSA}_{\text{DMS lifetime}}$ across all simulations (Fig. 7a). This

change is partially offset by a decrease in MSA/bioSO$_4$ in the Cala (GC13) and J. Chen (GC12) simulations, but the other

simulations amplify the increase by up to +3%.

Figure 7b shows that the modeled decrease in DMS lifetime between 1750 and 1979 contributes a 10–32% decrease in

Summit MSA deposition across all simulations. This $\Delta\text{MSA}_{\text{DMS lifetime}}$ is offset by a modeled increase in MSA/bioSO$_4$ in the

Tashmim (GC12) and Q. Chen (GC12) simulations, causing a net positive trend modeled MSA, which is in contrast to the

$57 \pm 19\%$ decrease in MSA over this time period in the Summit ice core. However, in the Cala (GC13), Tashmim (GC13)

and J. Chen (GC12) simulations, the decrease in MSA/bioSO$_4$ partitioning drives an additional decrease in MSA of 4–11%,




**Figure 7.** Ice core and modeled changes in MSA (top), bioSO$_4$ (middle) and MSA/bioSO$_4$ (bottom) at Denali (left) and Summit (right). Gray bars show the modeled change in MSA, bioSO$_4$, and MSA/bioSO$_4$ in the ice core grid cell due to change in DMS lifetime (eq. 1). Pink bars show the modeled change in MSA, bioSO$_4$, and MSA/bioSO$_4$ in the ice core grid cell due to change in MSA/bioSO$_4$ partitioning (eq. 2). Blue triangles show the net change in MSA, bioSO$_4$, and MSA/bioSO$_4$, and error bars are the range in net change in the surrounding grid cells. Ice core observations are shown as large blue triangles. This figure shows changes from 1750 to 1979, and similar changes between 1750 and 2007 are shown in Fig. S9.

which amplifies $\Delta$MSA$_{\text{DMS lifetime}}$ and qualitatively aligns with the Summit ice core MSA. In Figure 7d, $\Delta$bioSO$_{4\text{DMS lifetime}}$ ranges from $-10$ to $-32\%$, which is partially offset by the decrease in MSA/bioSO$_4$ in the Cala (GC13), Tashmim (GC13), and J. Chen (GC12) simulations, but amplified by the increase in MSA/bioSO$_4$ in the Tashmim (GC12) and Q. Chen (GC12) simulations (Fig. 7d). A larger increase in MSA/bioSO$_4$ of 50–90% would be needed to reproduce the observed $20 \pm 11\%$



increase in Summit ice core bioSO$_4$. However, while the Summit ice core MSA/bioSO$_4$ changes by $-64 \pm 37\%$ between the preindustrial and the MSA minimum (1969–1995), no model simulation reproduces a decrease of this magnitude (Fig. 7f). The model simulations show a wide range in MSA/bioSO$_4$ change, including of $-18 \pm 10\%$ in the Cala (GC13) simulation,

$-7 \pm 7\%$ in the Tashmim (GC13) simulation, $46 \pm 17\%$ in the Tashmim (GC12) simulation, $36 \pm 9\%$ in the Q. Chen (GC12) simulation, and $17 \pm 8\%$ in the J. Chen (GC12) simulation.

In summary, the overall similarity between box model results (Fig. 4) and GEOS-Chem $\Delta$MSA$_{\text{partitioning}}$, $\Delta$bioSO$_{4\text{partitioning}}$, and MSA/bioSO$_4$ (Fig. 7) suggest that discrepancies between GEOS-Chem and the box model are primarily driven by $\Delta$MSA$_{\text{DMS lifetime}}$. Additionally, overall better alignment between model simulations using GC13 and ice core observations suggest two takeaways.

First, better model-observation comparison in GC13 versus GC12 suggests that oxidant trends and concentrations over the industrial era could be more accurate in GC13 than GC12. Second, $\Delta$MSA$_{\text{DMS lifetime}}$ and $\Delta$bioSO$_{4\text{DMS lifetime}}$ in GEOS-Chem lead to misalignment with both ice core observations (Fig. 7) and box model results (Fig. 4), suggesting that under- or over-efficient transport and deposition could contribute to model-observation discrepancies in GEOS-Chem simulations, especially at Denali.

## 3.4    Comparison between model simulations and in situ observations

Figure 8 shows that simulations in GC13 better reproduce monthly surface MSA concentrations at four Arctic sites compared to GC12. The four sites include Alert, Canada (82°N, 62°W; 1980 to 2019), Ny Ålesund/Zeppelin, Svalbard (79°N, 12°E; 1990 to 2004), Utqiaġvik/Barrow, Alaska (71°N, 157°W; 1997 to 2022), and Qaanaak/Thule, Greenland (77°N, 69°W; 2010 to 2020) (Becagli et al., 2019; Quinn et al., 2009; Schmale et al., 2022; Sharma et al., 2019).

Model simulations using GC12 consistently overestimate surface MSA concentrations at Arctic sites by a factor of 2–80 during the months of highest MSA concentrations in the spring and summer (Figure 8). With an updated wet deposition scheme resulting in reduced atmospheric concentrations of soluble species (Luo et al., 2019, 2020), GC13 simulations with the Cala and Tashmim mechanisms overestimate spring/summer MSA concentrations by a smaller factor of 0–20 at Utqiaġvik, Qanaak, and Ny Ålesund (Fig. 8e), and underestimate spring/summer MSA concentrations at Alert or in winter/fall months

at other stations by up to a factor of 8. Winter MSA concentrations at Arctic stations are close to zero in observations and in GC13 simulations, but overestimated by up to 0.05 µg m$^{-3}$ at all stations in simulations using GC12. GEOS-Chem model simulations using the updated wet deposition scheme in GC13 (Luo et al., 2019, 2020) has been shown to better represent surface observations of aerosols in prior studies (Dutta and Heald, 2023; Gao et al., 2022). The global annual surface mean MSA and bioSO$_4$ concentrations in each simulation are shown in Figure S10.

Figure 9 compares observed and modeled DMS mixing ratio at four island or coastal sites. The four stations are Crete Island (CI; 35°N, 26°E; 1997–1999; Kouvarakis and Mihalopoulos, 2002), Amsterdam Island (AI; 38°S, 77°E; 1987–2006; Castebrunet et al., 2009) Cape Grim (CG; 40°S, 144° E; 1989–1992; Ayers et al., 1995), and Dumont D'Urville (DU; 66°S, 140°E; 1998–2006; Castebrunet et al.,2009). At Crete Island, observed DMS concentrations peak in July through October, but modeled DMS concentrations peak in May to June (Fig. 9b). The magnitude of the peak DMS mixing ratio (96–121 ppt)

in June across all simulations is similar to the peak observed mixing ratio (111-104 ppt) in July. Model mechanisms that do



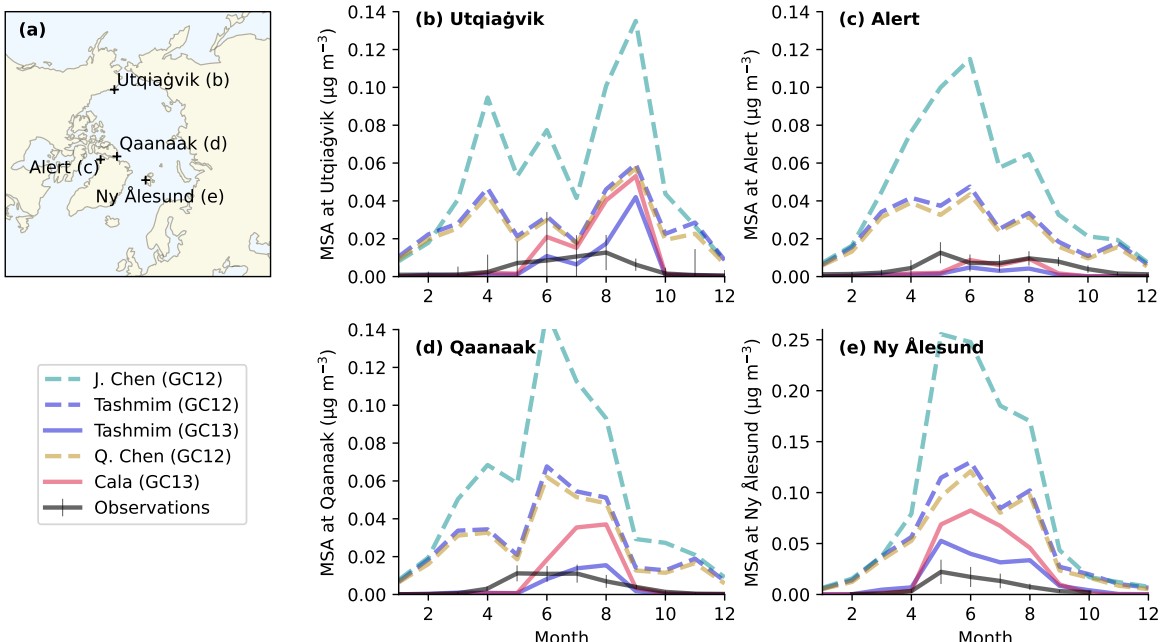

**Figure 8.** Observed monthly mean surface MSA concentrations (black lines in b-e) at four Arctic sites (a) compared to modeled MSA concentrations in several DMS oxidation mechanisms (colored lines in b-e). The four Arctic Sites include b) Utqiaġvik/Barrow, Alaska, c) Alert, Canada, d) Qaanaak/Thule, Greenland, and e) Ny Ålesund/Zeppelin, Svalbard. The error bars show the 25th to 75th percentiles and the solid black line is the monthly surface MSA concentration following Becagli et al. (2019). The simulations include the four mechanisms in Table 1, including the J. Chen mechanism in GC12 (dashed turquoise line), the Tashmim mechanism in GC12 (dashed purple line), the Tashmim mechanism in GC13 (solid purple line), the Q. Chen mechanism in GC12 (dashed yellow line), and the Cala mechanism in GC13 (solid pink line). DMS emissions are the same in all simulations.

not include DMS + BrO (Cala, J. Chen) do not reproduce the observed seasonality in DMS mixing ratio in Southern Ocean sites, similar to findings from Chen et al. (2018) (Figure 9c-e). Observed DMS missing ratio is at a maximum in in austral summer (DJF) at Amsterdam Island (Fig. 9c), Cape Grim (Fig. 9e), and Dumont D'Urville (Fig. 9e). This austral summer peak is reproduced by mechanisms that include DMS + BrO (Tashmim, Q. Chen), but these simulations underestimate summer
DMS mixing ratio by 50–70% during December at Cape Grim and Amsterdam Island and overestimate summer DMS mixing ratio by up to a factor of 3 at Dumont D'Urville. The mechanisms that do not include DMS + BrO (Cala, J. Chen) show a winter peak in DMS in the month of July at Dumont D'Urville (Fig. 9e) and variable DMS mixing ratios without a distinct seasonality at Amsterdam Island (Fig. 9c) and Cape Grim (Fig. 9d). In all three Southern Ocean sites, winter (JJA) DMS mixing ratio is overestimated by 20–180% in the J. Chen (GC12), Cala (GC13), Tashmim (GC12), and Q. Chen (GC12) simulation.
The discrepancy between observed and modeled DMS concentrations and seasonality may reflect inaccurate magnitude and seasonality in DMS emissions, missing or mischaracterized DMS oxidation chemistry, or both.



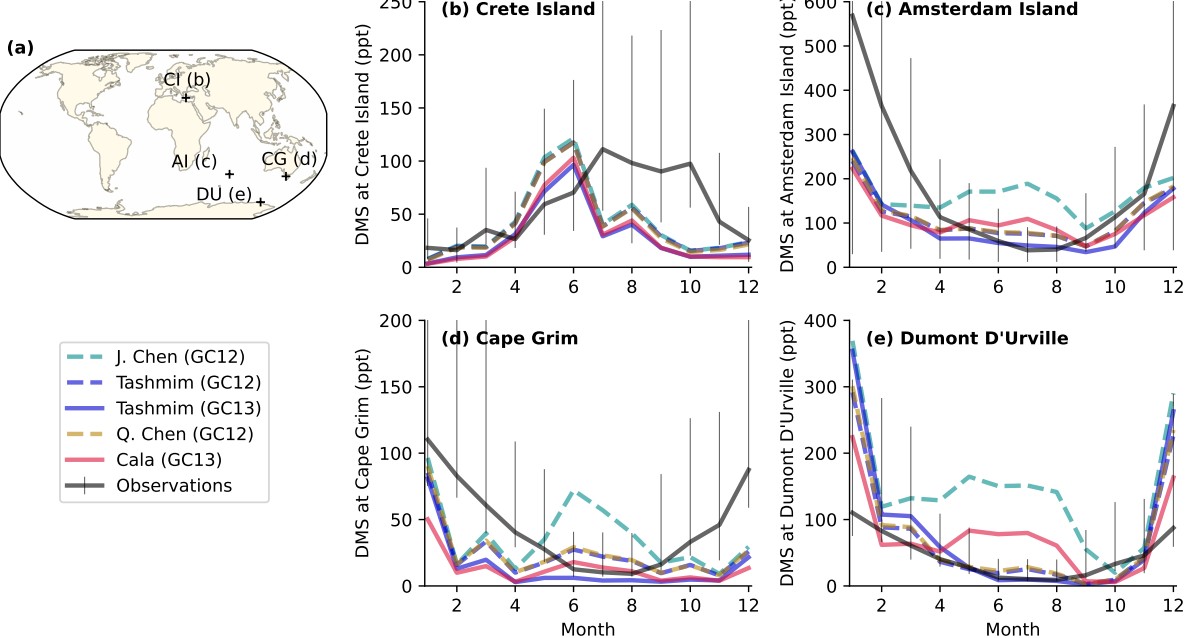

**Figure 9.** Monthly DMS mixing ratios in model simulations (colored lines) compared to long-term observations (black lines) at four sites including b) Crete Island, (CI; 35°N, 26°E), c) Amsterdam Island (AI; 38°S, 77°E), d) Cape Grim (CG; 40°S, 144°E), and e) Dumont D'Urville (DU; 66°S, 140°E). The error bars show the 25th to 75th percentiles and the solid black line is the monthly surface concentration anomaly of MSA following Chen et al. (2018). The simulations include the four mechanisms in Table 1, including the J. Chen mechanism in GC12 (dashes turquoise line), the Tashmim mechanism in GC12 (dashed purple line), the Tashmim mechanism in GC13 (solid purple line), the Q. Chen mechanism in GC12 (dashed yellow line), and the Cala mechanism in GC13 (solid pink line). DMS emissions are the same in all simulations.

Figure 10 compares modeled and observed ratios of MSA to non-sea salt sulfate (nssSO$_4^{2-}$) at 23 stations around the globe, and shows that all model simulations overestimate MSA/nssSO$_4^{2-}$ relative to observations at most sites. Most of the data are obtained from Gondwe et al. (2004), except for Crete Island from Kouvarakis and Mihalopoulos (2002) and Amsterdam

Island, Palmer, Kohnen, and Dome C from Casterbrunet et al. (2009). We compute the normalized mean bias (N$_{MB}$) for each simulation following Chen et al. (2018): N$_{MB} = \frac{\sum_{i=1}^{23}(M_i - O_i)}{\sum_{i=1}^{23} O_i}$, where M$_i$ is the modeled MSA/nssSO$_4^{2-}$ in the surface grid cell of each station and O$_i$ is the observed MSA/nssSO$_4^{2-}$. (N$_{MB}$) ranges from 155% to 692%, indicating a large overestimation in MSA/nssSO$_4^{2-}$ by all simulations (Fig. 10). The overestimate is largest in the Southern Hemisphere stations on the Antarctic Coast, where there is negligible influence from anthropogenic emissions on nssSO$_4^{2-}$. Observed MSA/nssSO$_4^{2-}$ ranges from

0.005 (Crete Island) to 0.35 (Palmer). Modeled MSA/nssSO$_4^{2-}$ ranges from 0.006–3.02 in model simulations. The maximum MSA/nssSO$_4^{2-}$ in each modeled mechanism is 0.61 to 3.02, a factor of 1.8–8.4 higher than the observed maximum. We note that



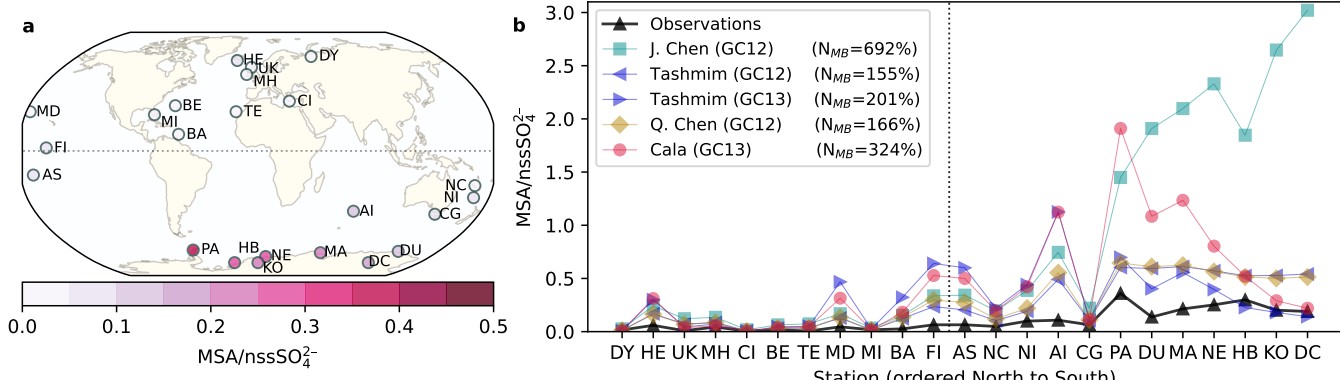

**Figure 10.** Comparison between modeled and observed $MSA/nssSO_4^{2-}$. a) Observed annual mean $MSA/nssSO_4^{2-}$ at 23 sites around the world. b) Comparison between observed annual mean $MSA/nssSO_4^{2-}$ and modeled $MSA/nssSO_4^{2-}$ in the J. Chen (GC12) simulation (turquoise square), the Tashmim (GC12) simulation (left-pointing purple triangle), the Tashmim (GC13) simulation (right-pointing purple triangle), the Q. Chen (GC12) simulation (yellow diamond), and the Cala (GC13) simulation (pink circle) using the same 2007 meteorology and emissions in all simulations. Stations include Dye (DY; 66˚N, 53˚E), Heimaey (HE; 63˚N, 20˚W), United Kingdom (UK; 58˚N, 20˚W), Mace Head (MH; 53˚N, 10˚W), Crete Island (CI; 35˚N, 25˚E), Bermuda (BE; 32˚N, 65˚W), Tenerife (TE; 28˚N, 17˚W), Midway Island (MD; 28˚N, 177˚W), Miami (MI; 26˚N, 80˚W), Barbados (BA; 13˚N, 60˚W), Fanning Island (FI; 4˚N, 159˚W), American Samoa (AS; 14˚S, 170˚W), New Caledonia (NC; 21˚S, 166˚E), Norfolk Island (NI; 29˚S, 168˚E), Amsterdam Island (AI; 38˚S, 77˚E), Cape Grim (40˚S, 144˚E), Palmer (PA; 65˚S, 64˚W), Dumont D'Urville (DU; 66˚S, 140˚E), Mawson (MA; 67˚S, 63˚E), Neumayer (NE; 70˚S, 8˚W), Halley Bay (HB; 75˚S, 26˚W), Kohnen (KO; 75˚S, 0˚E), and Dome C (DC; 75˚S, 123˚E). The legend also shows the normalized mean bias ($N_{MB}$) for each simulation. The dashed gray line shows the delineation between northern and southern hemisphere sites.

discrepancies between observed and modeled $MSA/nssSO_4^{2-}$ could occur due to mischaracterized DMS oxidation chemistry, $SO_2$ oxidation chemistry, DMS emissions, anthropogenic $SO_2$ emissions, and/or natural sulfur emissions.

# 4    Conclusions

We investigate DMS oxidation chemistry over the industrial era by comparing model simulations with four different DMS chemical oxidation mechanisms to ice core and in situ observations of DMS, MSA, $bioSO_4$, and $MSA/nssSO_4^{2-}$. Jongebloed et al. (2023a) and Chalif et al. (2024) hypothesize that a pollution-driven increase in nitrate radical in the Summit and Denali ice core source regions drove the observed industrial-era decline in ice core MSA and concurrent increase in $bioSO_4$. We show that GEOS-Chem and box model simulations can reproduce trends in DMS oxidation products at Summit, but different oxidation

mechanisms and model versions lead to a wide range in results, and only box model simulations capture trends at Denali. In agreement with the hypothesized $NO_3$-driven MSA decline from Jongebloed et al. (2023a) and Chalif et al. (2024), we find that $DMS + NO_3$ increased over the industrial era in both the North Atlantic and North Pacific regions, favoring the production of $bioSO_4$ in all simulations and driving a decrease in MSA, which aligns with ice core observations. In simulations that




include the reaction of DMS with BrO, the industrial-era increase in BrO drives an increase the production of MSA and offsets
the $NO_3$-driven decrease, which results in a discrepancy between modeled and observed trends in MSA in some simulations.
DMS + BrO is needed to capture the seasonality of atmospheric concentrations of DMS, but a potential overestimate in
MSA production in simulations with DMS + BrO could result from overestimated BrO concentrations, underestimated $NO_3$
concentrations, overly efficient MSA production from the addition pathway, or other missing or misrepresented DMS oxidation
chemistry.

Substantially different results using the Tashmim mechanism in two different model versions (GC12 and GC13 in both
GEOS-Chem and box modeling) show that inaccurate oxidant concentrations may contribute to model-observation discrepan-
cies in simulations that cannot reproduce ice core trends. The sensitivity of our results to oxidant concentrations suggests that
improving our understanding of oxidant changes is critical to improving comparison between modeled and observed DMS ox-
idation products. GEOS-Chem and box model simulations using GC13 better align with ice core trends, suggesting that GC13
may more accurately represent trends and concentrations in oxidants over the industrial era compared to GC12. Interestingly,
the trends, seasonality, and surface concentrations in MSA, $bioSO_4$, DMS, and $MSA/nssSO_4^{2-}$ are similar in the Tashmim
and Q. Chen mechanisms when using the same model version. The Q. Chen mechanism includes DMS + OH, $NO_3$, BrO, Cl,
and $O_3$, and intermediates such as DMSO and MSIA, but does not explicitly account for the formation of HPMTF and other
short-lived isomerization pathway intermediates, suggesting that a simplified mechanism for DMS chemistry may be sufficient
in modeling the abundance, seasonality, and trends DMS-derived aerosols and their effects on global radiative forcing.

The discrepancies between observed and modeled trends in MSA and $bioSO_4$ in some simulations might imply missing or
misrepresented DMS oxidation chemistry. Recent studies investigating gas-phase DMS chemistry have discovered important
pathways of MSA and sulfate production through intermediates such as HPMTF, MSP, and $CH_3SO_2$. We suggest that future
studies should investigate aqueous-phase chemistry, which produces 82–99% of MSA and $bioSO_4$ in our simulations. Addi-
tionally, future studies analyzing the oxygen isotopes of MSA might indicate missing or misrepresented chemistry in the DMS
oxidation mechanism by quantifying the importance of different oxidation pathways, similar to previous studies quantifying
sulfate formation through $\Delta^{17}O(SO_4^{2-})$ (Hattori et al., 2021, 2024; Sofen et al., 2011). Inclusion of methanethiol may improve
model-observation comparison because methanethiol favors $bioSO_4$ production over MSA (Novak et al., 2022). Finally, it
is possible that uncertainty in reaction rates for key reactions (e.g., DMS + $NO_3$) could contribute to discrepancies between
modeled and observed trends in DMS oxidation products.

In general, interpretation of ice core or in situ observations of short-lived oxidized species, such as MSA and sulfate, should
consider how changes in the lifetimes of precursors and in DMS oxidation pathways can influence long-term trends. For ex-
ample, in regions influenced by pollution or other factors that affect oxidant concentrations, trends in MSA should be assumed
to at least partially reflect changing oxidation chemistry. Currently, model simulations alone cannot be used to estimate the
potential influence of changing atmospheric chemistry on long-term trends in MSA. Instead, measurements of sulfur isotopes
that provide estimates of total biogenic sulfur (MSA + $bioSO_4$) and the ratio of oxidation products ($MSA/bioSO_4$) can indicate
whether and how much atmospheric chemistry has influenced DMS oxidation and trends in MSA and $bioSO_4$.



DMS is a major source of aerosol and cloud condensation nuclei that influence global climate and is an increasingly large fraction of atmospheric sulfate as anthropogenic pollution emissions decrease (Jongebloed et al., 2023a). The four mechanisms

of DMS oxidation tested in this study simulate different magnitudes of MSA and bioSO$_4$, and different fractions of MSA and sulfate produced in the gas vs. aqueous phase, which has implications for new particle formation and aerosol-cloud interactions. Understanding and accurately modeling DMS oxidation is critical for understanding past and future climate, especially in light of proposed marine cloud brightening efforts to offset warming caused by greenhouse gases and potential future decreases in anthropogenic aerosol radiative forcing.

*Code and data availability.* Ice core data were obtained from the referenced papers Jongebloed et al. (2023a, b, c); Chalif et al. (2024) and are available in the NSF Arctic Data Center at https://doi.org/10.18739/A2WW7717K, https://doi.org/10.18739/A2N873162, https://doi.org/10.18739/A26T0GX7K, and https//doi.org/10.18739/A2Q814T9K. In situ data were obtained from the referenced papers (Ayers et al., 1995; Becagli et al., 2019; Gondwe et al., 2004; Kouvarakis and Mihalopoulos, 2002; Quinn et al., 2009; Schmale et al., 2022). GEOS-Chem versions 12.9.3 and 13.2.1 are available online (https://zenodo.org/records/3974569 and https://doi.org/10.5281/zenodo.5500717).

*Author contributions.* U.A.J. designed the research study, ran the simulations, performed the analysis, and wrote the manuscript. B.A. initiated and supervised the study. J.I.C. contributed to study design and J.I.C., E.C.O., and B.G.K. contributed to analysis. B.A., L.T., W.P., K.H.B., and Q.C. contributed to model development. U.A.J., B.A., and J.C.-D. contributed to Summit ice core data collection and analysis. J.I.C., E.C.O., B.G.K., D.A.W., K.J.K., D.G.F., and C.P.W. contributed to Denali ice core data collection and analysis. All authors contributed to interpretation and writing.

*Competing interests.* At least one of the (co-)authors is a member of the editorial board of Atmospheric Chemistry and Physics.

*Acknowledgements.* U.A.J. and B.A. acknowledge awards PLR 1904148, PLR 2230350, and AGS 2202287. Q.C. acknowledges the Hong Kong Research Grants Council (Grant Nos. 15223221 and 15219722). D.A.W. acknowledges awards AGS 1204035 and OPP 2002470. K.J.K. acknowledges AGS 0713974, 1203838, 1502783, 1806422, 2002483. E.C.O. and J.I.C. acknowledge AGS 1204035 and C.P.W. acknowledges AGS 1203863.



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
