# Peer review of "Dimethyl sulfide chemistry over the industrial era: comparison of key oxidation mechanisms and long-term observations"

_EGUsphere, 2024_

## Author Comment (AC1)

**Response to Referee 1**

Jongebloed and coauthors' responses in blue

This study compares a series of global and box model simulations with varying DMS chemistry mechanisms to the long-term trend in ice core trends in MSA at Denali and Summit (where bio-sulfate was also estimated from measurements). They find that no scheme is able to reproduce the observed trends. They explore the role of oxidants, lifetime, and indirectly, deposition schemes on their results. They also compare simulated present-day concentrations with observed MSA, DMS, and MSA/sulfate ratios. The study is thorough and detailed. While the authors were not able to fully explain the observed trend with any of their model schemes, the study explores a range of sensitivities and remains a valuable contribution to the literature. I include comments below largely to improve the readability of the manuscript.

We thank Referee 1 for their detailed review of this paper. We believe our changes made based on their comments have made the paper more accurate and easier to digest. Please find our responses to major and minor comments below.

Major comments:
1.  The discussion of 3.2 is challenging to follow with all the various model versions and time intervals. And given that no model is able to fully reproduce the observed trends, I would suggest that the authors consider separating their discussion into two time horizons: 1750-1979 and then 1979-2007.

    We have updated 3.2 to examine each time horizon (1750-1979 vs. 1979-2007). We also shortened the text in this section. We simplified Figure 4 to make it more digestible. In addition, we have removed the panels from Figures 4 and 7 that do not contain observations (i.e., bioSO4 and MSA/bioSO4 at Denali) to simplify the comparison.

2.  Line 9-10; line 269, line 424, line 424: These statements are not correct. Based on Figure 4, no model can fully reproduce the observed trends at Dinali or Summit. There are some schemes that have success in reproducing some of the trend (e.g. 1750 to 1979, but not 1979 to 2007), or the overall tendency (if not magnitude) in some species, but not others. More broadly, the authors should be clear that the models all do simulate an increasing role for NO3 oxidation of DMS, but this does not produce the observed decline in MSA and therefore does not directly support their hypothesis (lines 426-428 are incorrect – no model reproduces the observed decrease in MSA at Denali and only some models reproduce the decrease in MSA at Summit from 1750-1979, but do not capture the increase from 1979-2007).

    We agree that the ability of models to capture ice core trends is overstated in the previous draft of this paper. We have made changes throughout the text:
    - Line 9-10: We have changed the abstract to clarify that some box model simulations can capture some ice core trends, and GEOS-Chem simulations do not capture trends. We have also clarified that these results support the increased role of the nitrate radical in DMS oxidation over the industrial era, but that

competing factors (increased BrO oxidation, decreased DMS lifetime) offset the increased bioSO4 production from the nitrate radical in some simulations.

- Line 269: Similar to the abstract, we clarify that some box model trends qualitatively capture trends in the Summit ice core, but no box or GEOS-Chem simulations capture trends at Denali, and no GEOS-Chem simulations capture all trends at Summit. (now line 286)
- Line 424: Updated: "We show that some box model simulations can qualitatively reproduce trends in DMS oxidation products at Summit, but GEOS-Chem simulations cannot reproduce Summit trends in MSA or bioSO4, and no box model or GEOS-Chem simulation can reproduce trends at Denali. Additionally, different oxidation mechanisms and model versions lead to a wide range in results." (now line 433)

3. Additional text that mischaracterizes the results:
   o Line 210-211: nitrate did not plateau from 1979-2007 in v13.2.1
     We have updated the sentence to say "increased further or plateaued"
   o Lines 227-228: the box model using Cala and Tashmim mechanisms and GC13 only reproduce the 1750 to 1979 trend, they do not represent the increase from 1979 to 2007.
     Corrected by dividing section 3.2 into 1750–1979 and 1979–2007
   o Line 286-287: This statement is incorrect: despite the consistent increasing trend in NO3 in Figure 5, Figure 4 shows that, in none of the simulations does bioSO4 increase monotonically, nor does MSA decrease monotonically over time.
     This line is based on Figure 1, which shows that DMS + NO3 proceeds through the abstraction + isomerization pathway, which mainly produces bioSO4. We have rephrased the sentence to clarify:
       This increase in DMS + NO3 drives an increase in DMS oxidation through the isomerization pathway, which favors the production of bioSO4
   o Line 369, line 439: It's not clear from Figure 4 that GC13 outperforms GC12 in comparison to the ice core observations. For MSA, GC13 is superior to GC12 at Summit, but worse at Denali. The performance for bioSO4 at Summit is poor for both. These statements should be corrected.
     The statements have been removed.

Minor comments:
1. All figure captions: please specify what "model results" are being compared. Concentrations? Deposition?

   We updated the figure caption to specify model result wherever it was not previously mentioned:
   1. Figure 4 caption updated to specify deposition
   2. Figure 7 caption updated to specify deposition.
   3. Figure 10 now says comparison between "annual-mean surface MSA/nssSO4 concentration ratio."
2. Line 14 and 449: while much of the chemistry goes via an aqueous pathway, this sentence seems to suggest that it is that part of the chemical processing which is uncertain. While this

may be true for aqueous phase MSA production, it seems more likely that rather than the aqueous conversion of SO2 to sulfate, it is the gas-phase chemistry that precedes this step that is uncertain. Perhaps this statements should be modified to be more specific to the aqueous pathways of interest?

> This is a good point: the aqueous-phase reaction of SO2 → sulfate is less uncertain than some of the gas-phase reactions. However, we were also referring to the uncertainty of the aqueous-phase reaction of HPMTF to form sulfate, which is responsible for the majority (>75%) of the bioSO4 formation in simulations that include HPMTF chemistry (i.e., all but the Q. Chen mechanism), but the chemical reaction(s) forming sulfate from HPMTF in the aqueous phase are currently unknown. We have left the abstract sentence to remain as it is due to space constraints (250 words), but we have updated the conclusion (lines 461–463) to specify the uncertainty we refer to in this sentence.

3. Lines 143-144: Not including MSA+OH does not seem very well justified. Is there any literature to support this decision beyond it being "overly efficient" in the Tashmim mechanism? Perhaps the authors could discuss the uncertainty in the rate? MSA is overestimated in many of the simulations shown in Figure 8 and it seems like this loss would ameliorate some of these comparisons. It would be nice to see some further discussion of this.

> We agree that this could use further discussion. We have added a reference to Mungall et al. (2018), which estimate that the lifetime of MSA against OH-oxidation in cloudwater should be about one year, which is much longer than the atmospheric lifetime of MSA, indicating that this reaction is likely inefficient in the atmosphere. Fung et al. (2022) use the same aqueous-phase reaction as the Tashmim mechanism and estimate that 76% of MSA is lost through this reaction, but they do not examine the seasonality of MSA in their simulations, so we hypothesize that this implementation is overly efficient in their simulations as well. Mungall et al. (2018) find that MSA + OH in aerosol might have a lifetime of 5–7 days and should be more efficient than MSA + OH in cloudwater, but we did not include this reaction in our simulations.

> We have also added to the concluding paragraph a reference to the uncertainty in MSA + OH (line 468) as a source of uncertainty potentially leading to model-observation discrepancies

4. Line 174-183: what are the uncertainties on all of these measurements?

> Uncertainty in measurements is shown as error bars on Figure 2 and now described in Section 2.4 (line 196).

5. Lines 189-190: what is the uncertainty associated with this comparison of grid cell average deposition at 4x5? Can the authors comments on the possible impact of uncertainties in transport, deposition, and inability to reproduce gradients at this very coarse horizontal resolution?

We agree that it is possible that some of the model-observation discrepancies between GEOS-Chem deposition and ice core trends are affected by low model resolution. We estimate the uncertainty in deposition by taking the average over several surrounding grid cells (which is how we calculate the error bars in Figures 4 and 7). We have added a sentence to the summary paragraph noting that the low model resolution may be impacting our model-observation comparison: "The GEOS-Chem modeled change in DMS lifetime may also be sensitive to the transport and deposition efficiencies of MSA and sulfate (Figure 6), which may not be represented well in simulations at low $4° \times 5°$ model resolution." (lines 381–365)

6. Figure 6b: The large percentages here are largely over low deposition regions (differences of small numbers). Presumably at lower deposition, uncertainties may be larger? Can the authors comment on this.

   Yes, the change in deposition is large in areas where the deposition of DMS-derived species is low (high latitudes, continents). This is part of why using ice cores to examine past trends is difficult: these are high latitude, high altitude regions with relatively low deposition of important atmospheric tracers such as MSA and sulfate. We added a sentence: "The percentage change is especially large in regions with low deposition (e.g., Greenland)" (line 327).

7. Figure 8, 9, 10: The authors might choose colours to better distinguish GC12 and GC 13 (e.g. warm colours for GC13, cold colours for GC12) to improve ease of interpretation.

   We use dashed vs. solid lines for GC13 vs. GC12, and colors to distinguish mechanisms. So two of the simulations (Tashmim GC12 and Tashmim GC13) need to be the same color. We think it would be more difficult to interpret figures where different mechanisms are shown with different line styles and model versions are shown in two colors.

Thank you for these helpful suggestions, we believe these comments have helped make the paper more accurate and easier to read.

**Response to Referee 2**

Jongebloed and coauthors' responses in blue

This study has used two versions of GEOS-Chem with four different DMS oxidation mechanisms implemented (in total 5 different simulations) to investigate how the oxidation mechanisms influences the long-term trend in DMS derived sulphate and MSA and compare the results to ice core observations. The trends differ, depending on the mechanisms included and the model version used (with different atmospheric oxidant concentrations), highlighting the importance of the sulphur chemistry. None of the simulations could reproduce both the long-term trend and the seasonality in in situ measurements. For aerosol-cloud interaction, the natural aerosols background level is important, and hence better understanding of the natural sulphur cycle is important.

> We thank Referee 2 for their detailed review and helpful comments. We have made changes based on these suggestions that we believe have improved the paper. We have responded to each suggestion below.

The study is well defined and highlights important issues in atmospheric chemistry modelling. Some improvement to the method sections to make the set up clearer is needed. And the flow in the results section could be improved. The results section is sometimes hard to follow, but the authors have added a summary section at the end of each section which is good. The conclusion sections put the results in a broader context. One more issue that is worth mentioned the role of DMS on the effect of the IMO2020 ship emission regulation, as shown in Jin et al. (2018). Below are my specific comments to the manuscript:

> We agree that the results section needed improved flow. We have re-organized section 3.2 into two different time periods to make the results section easier to follow. We have also moved the Antarctic comparison into its own sub-section. Finally, we combined the first two paragraphs describing Figure 5 to shorten section 3.3.

> We have not explored how this work has implications for radiative forcing of anthropogenic sulfate (e.g., ship emissions) because our model does not include aerosol-cloud interactions, so it would be difficult to quantify the effect of DMS chemistry on forcing adjustments from IMO2020 ship regulations (Jin et al., 2018). However, it is an important possible implication of modeling differences in DMS oxidation chemistry so we have added this reference to the introduction (line 23).

L95: As I was very curious about the difference between the two GEOS-chem versions, I would have rearranged the first paragraph in section 2.1. First present GEOS-chem and at the end the two different models and how they differ. Do you know what else is different than wet deposition? The natural emissions are also identical in the two model versions.

> You are correct that the natural emissions are identical in the two model versions. The wet deposition scheme is the main difference between two versions, which is discussed toward the end of the first paragraph in section 2.1. It is unclear why the oxidant

concentrations are so different in each model version. It could be a combination of a structural change in the model between version 12 and 13 combined with minor updates to chemistry and changes to wet deposition, which are all documented on the GEOS-Chem website. Identifying the cause of the differences is beyond the scope of this paper.

L106 and several other places you refer to the model versions 13.2.1 or 12.9.3. Stick to GC12 and GC13 as they are defined.

We have changed all instances of 12.9.3 to GC12 and 13.2.1 to GC13.

L110: Clearly define the abbreviations for the chemistry schemes before they are used. It may be useful to refer to Table 1 here and maybe skip the details of the chemistry here and leave that for section 2.2. And why is only one scheme used in both models?

We have removed the abbreviated mechanism names from this paragraph. We only implemented one mechanism (the Tashmim mechanism) into two model versions due to the computational expense of running many more mechanisms in other model versions.

L116: Ice cover is also equal in all simulations? Ice cover was mentioned in the Introduction.

We have added that sea ice cover is the same in all simulations in Section 2.1 (line 109)

L122: Can you add the total DMS emissions in the model simulations here? And how it compares to other studies.

We have added emissions of DMS from our simulations and compared them to other estimates in the first paragraph in section 2.1 (line 102).

L130: Add also range of absolute numbers in emissions? And what is the source of these emissions?

We added the range in emissions used across various global models from Wang et al. (2020) to line 102 and also line 125. The source of the emissions in our simulations is Lana et al. (2011) (line 102).

L135: Table 2 with the time periods simulated should be referred to in the previous section, and keep this section only for describing the oxidation mechanisms.

We added references to Table 1 (line 96) and Table 2 (line 110) in the first paragraph of Section 2.1 and removed the reference to Table 2 in this sentence.

Figure 1: In the figure caption, can the abbreviation be used instead of references? And more clearly state what part of the figures that are not included in the different schemes? The figures does not tell about the differences between the schemes, so a reference to the supplementary figures at the end of the table caption could be good. And regarding the supplementary figures: Is there no abstraction branch in J. Chen and Cala? Could the figures have a similar layout, so it is

easier to visually grasp the differences? Can the numbers below the arrows in Fig. 1 be added to these supplementary figures?

> We have added the abbreviated names to Figure 1 and removed the excessive references. We also added a reference to the supplementary figures to depict what reactions are included in each mechanism. We have updated the supplementary Figures S4 and S5 to be a similar layout to Figures S2 and S3. We have added numbers below/above arrows where possible.

L165: "isolate the impacts of changing oxidant concentrations on trends in MSA and bioSO4" In the full simulations, what else are impacting? Add advantages of using box-model compared to the full model.

> We added this sentence to section 2.1 line 177: "The box model does not include emissions, transport, or deposition, therefore allowing the effects of changing oxidant concentrations to be isolated from other processes." We also discuss the advantages of the box model further in the results.

L197: For clarity, state what chemistry scheme is used in these simulations as listed in Table 1. Does the different schemes have any impact at all on these values? I guess not. Maybe the default scheme in the two model versions can be mentioned in the GEOS-chem section.

> We showed the Tashmim mechanism in Figure 3, and clarified this in line 214. We also added this sentence: "Most oxidants (e.g., $O_3$, Cl, OH, and $NO_3$) vary by <1% between mechanisms within the same model version. BrO, however, is up to 14% different between mechanisms that include DMS + BrO versus mechanisms that do not include DMS + BrO, suggesting that DMS oxidation is an important sink for BrO."

L206: Can you describe the Cl trend in Zhai et al?

> We decided to remove the reference to Zhai et al. (2021) here because the main reason we do not show Cl is because it is a minor oxidant of DMS. In Zhai et al. (2021) and our GC12 model, Cl decreases by 32% between 1750 and 1979 and increases by 26% between 1979 and 2007. In our GC13 model, Cl increases by 20% between 1750 and 1979 and increases by an additional 10% between 1979 and 2007.

L218: Can you put your results in the context of the more recent multi model intercomparison AerChemMIP as well? (Griffiths et al., 2021)

> Thank you for pointing out the more recent multi model intercomparison studies. We have updated the text to compare our results to Griffiths et al. (2021) (line 236).

L224: Here as well, you can refer to more recent multi model studies (Stevenson et al., 2020) from AerChemMIP.

> We have updated the text to compare our results to Stevenson et al. (2020) (line 242).

L229 and L235: "some simulations" This is a bit vague. Maybe skip and write which capture the observed trend and which do not.

> We agree that this is vague. We have re-written this section to clarify the model-observation comparisons. First, we examine each time horizon in separate subsections of Section 3.2 (1750-1979 vs. 1979-2007). We also shortened the text in this section and simplified Figure 4. In addition, we have removed the panels from Figures 4 and 7 that do not contain observations (e.g., bioSO4 and MSA/bioSO4 at Denali) to simplify the comparison.

I also struggle a bit with the box model vs. the model results. Can why they differ (for components and sites) be explained? Near L370 transport and deposition is mentioned. Can this be brought in earlier in the text.

> We believe the differences between box model and GEOS-Chem results are best explained by considering the changes to DMS lifetime (controlled by oxidants and distance transported from emission region), which we discuss in more detail in section 3.3. In section 3.2, we add this sentence (line 255): "The reasons for the discrepancy between GEOS-Chem simulations, box model simulations, and Denali observations are explored in Section 3.3." And later (line 301): "The differences in results across box model simulations, GEOS-Chem model versions, and DMS oxidation mechanisms are investigated in Section 3.3."

L269: Can you bring in results from Denali in the summary as well?

> We added this sentence (line 288): "Neither box model nor GEOS-Chem simulations fully reproduce trends in MSA at Denali, but box model simulations can reproduce a decrease between 1750 and 1979 observed in the Denali ice core."

L286: Fig 5b -> Fig 5a?

> Yes, changed to 5a.

L305: In the section above, you have presented each of the figure panels separately. Possible to combine this presentation, and highlight differences.

> We agree that this section can be made more concise by combining the first two paragraphs of this section. We have combined these results into one paragraph starting in line 301.

L321: Add a description of the trend in the ice core records from the previous studies.

> We added this sentence (line 375): "Antarctic ice core studies find changes in MSA ranging from negligible (e.g., West Antarctica; Osman et al., 2017) to substantial (e.g., 20–30% in East Antarctica Curran et al., 2003). We find that the model simulates a

change in MSA, bioSO4, and MSA/bioSO4 of < ±10% at any ice core location (Fig. S8)." Note that one of the papers we cited was recently withdrawn (Nilsson et al., 2024), so we have removed this reference.

L367: "In summary, the overall similarity between box model results (Fig. 4) and GEOS-Chem.. (Fig. 7)" I can not see that you have discussed Fig. 4 or the box model results in the section above. Can you please help the reader describe this?

Yes, we have clarified this in the text. In line 247: "The changes in MSA and bioSO4 caused by a change in partitioning are qualitatively similar to the box model results. For example, in GEOS-Chem simulations using the Tashmim mechanism, $\Delta bioSO_{4partitioning}$ is positive in GC13 and negative in GC12 (Fig. 7c), which aligns qualitatively with the box model results (Fig. 4c). Additionally, the Cala (GC13) and J. Chen (GC12) simulations produce the largest decrease in MSA at Summit in both the box model and in $\Delta MSA_{partitioning}$"

Figure 8 and Figure 9: Add time period in the figure captions.

We added time periods for each site.

Figure 10: Add time period for the observations in the figure.

The time period sampled varies significantly by site, making it difficult to summarize concisely in a figure caption. We have added this note to the figure caption and refer the reader to the cited sources.

L421: The in situ observations is used for comparing the model results for present day and not over the industrial era. Consider rewriting. Maybe introduce the in situ in L431?

We implemented this suggestion (line 432).

L435: Remind the reader that these two model versions have different oxidant concentrations.

We implemented this suggestion (line 447).

L444: "but does not explicitly account for the formation of HPMTF and other short-lived isomerization pathway intermediates" add "as included in Tashmim" (if correct)

HPMTF and other intermediates are included in all mechanisms (Tashmim, J. Chen, and Cala) except for Q. Chen. We are leaving this sentence as is to avoid over-complicating it since it is already a long sentence.

L446: "some simulations" list which ones.

We have replaced "some" with "GEOS-Chem" because no simulations capture all observed trends in MSA and bioSO4 (line 458).

Table 2 and 3 in the supplement, burden has unit mass, but given in these tables as mass per year, please check. And how is the lifetime calculated?

> Yes, burden units should be mass, not mass per year. Lifetime is calculated by dividing burden by loss (dry + wet deposition) or dividing burden by production (they give the same answer). We have updated the table to correct the units and describe the calculation.

Technical comments:
L39-40: "with updated DMS oxidation chemistry" mentioned twice.
> Thank you for catching this.

L304: No Fig 5e.
> This sentence is removed (see suggestion above to combine the two paragraphs on Figure 5)

L332: "lifetime can offset or amplify a trend in that occurs due" delete in.
> Corrected

L397: "Observed DMS missing ratio is at a maximum" -> mixing I guess.
> Yes, thank you.

L429: "increase in BrO drives an increase the production of MSA" -> delete the.
> Corrected.

Figure 2: a), b), c) etc. missing in the figure.
> We have added a, b, c, d, etc. back into the figure.

Figure 4d: a triangle is shown.
> Thank you for catching this. We have changed it to a circle.

**References:**
Griffiths, P. T., Murray, L. T., Zeng, G., Shin, Y. M., Abraham, N. L., Archibald, A. T., Deushi, M., Emmons, L. K., Galbally, I. E., Hassler, B., Horowitz, L. W., Keeble, J., Liu, J., Moeini, O., Naik, V., O'Connor, F. M., Oshima, N., Tarasick, D., Tilmes, S., Turnock, S. T., Wild, O., Young, P. J., and Zanis, P.: Tropospheric ozone in CMIP6 simulations, Atmos. Chem. Phys., 21,4187-4218, 10.5194/acp-21-4187-2021, 2021.
Jin, Q., Grandey, B. S., Rothenberg, D., Avramov, A., and Wang, C.: Impacts on cloud radiative effects induced by coexisting aerosols converted from international shipping and maritime DMS emissions, Atmos. Chem. Phys., 18,16793-16808, 10.5194/acp-18-16793-2018, 2018.
Stevenson, D. S., Zhao, A., Naik, V., O'Connor, F. M., Tilmes, S., Zeng, G., Murray, L. T., Collins, W. J., Griffiths, P. T., Shim, S., Horowitz, L. W., Sentman, L. T., and Emmons, L.: Trends in global tropospheric hydroxyl radical and methane lifetime since 1850 from AerChemMIP, Atmos. Chem. Phys., 20,12905-12920, 10.5194/acp-20-12905-2020, 2020.

> Thank you for your detailed review. We believe the updates based on these suggestions have improved the readability and utility of the paper.